# Loss of the extracellular matrix protein Perlecan disrupts axonal and synaptic stability during *Drosophila* development

Ellen J Guss, Yulia Akbergenova, Karen L Cunningham, J Troy Littleton*

The Picower Institute for Learning and Memory, Department of Biology, Department of Brain and Cognitive Sciences, Massachusetts Institute of Technology, Cambridge, United States

**Abstract** Heparan sulfate proteoglycans (HSPGs) form essential components of the extracellular matrix (ECM) and basement membrane (BM) and have both structural and signaling roles. Perlecan is a secreted ECM-localized HSPG that contributes to tissue integrity and cell-cell communication. Although a core component of the ECM, the role of Perlecan in neuronal structure and function is less understood. Here, we identify a role for *Drosophila* Perlecan in the maintenance of larval motoneuron axonal and synaptic stability. Loss of Perlecan causes alterations in the axonal cytoskeleton, followed by axonal breakage and synaptic retraction of neuromuscular junctions. These phenotypes are not prevented by blocking Wallerian degeneration and are independent of Perlecan's role in Wingless signaling. Expression of Perlecan solely in motoneurons cannot rescue synaptic retraction phenotypes. Similarly, removing Perlecan specifically from neurons, glia, or muscle does not cause synaptic retraction, indicating the protein is secreted from multiple cell types and functions non-cell autonomously. Within the peripheral nervous system, Perlecan predominantly localizes to the neural lamella, a specialized ECM surrounding nerve bundles. Indeed, the neural lamella is disrupted in the absence of Perlecan, with axons occasionally exiting their usual boundary in the nerve bundle. In addition, entire nerve bundles degenerate in a temporally coordinated manner across individual hemi-segments throughout larval development. These observations indicate disruption of neural lamella ECM function triggers axonal destabilization and synaptic retraction of motoneurons, revealing a role for Perlecan in axonal and synaptic integrity during nervous system development.

*For correspondence:
troy@mit.edu

Competing interest: The authors declare that no competing interests exist.

## eLife assessment

This study presents **valuable** new insights into the role of the extracellular matrix component (ECM) Perlecan in axon integrity, with downstream consequences for the maintenance of synaptic structures. The evidence for Perlecan's role in this process is **solid**, although negative results for Perlecan's mechanism of action should be strengthened with the addition of appropriate controls centered on the relevant pathways and mechanisms involved as well as more careful analyses and interpretations. The authors provide **convincing** data identifying and describing the cellular sequence from ECM perturbations to axonal and synaptic degeneration, but additional data pinpointing the requirements of Perlecan for axonal maintenance would further improve the impact of this study.

## Introduction

Neurons require regulated polarization and transport of synaptic material to maintain their distinctive shape and electrical properties. Indeed, disruption of axonal transport is linked to numerous

neurodevelopmental and neurodegenerative disorders (*Cheng et al., 2022*; *De Vos et al., 2008*; *DiAntonio, 2019*; *Fernandopulle et al., 2021*; *Krench and Littleton, 2013*; *Luo and O'Leary, 2005*; *Mariano et al., 2018*; *Neukomm and Freeman, 2014*). Neuronal development depends upon multiple transmembrane and secreted proteins that facilitate intercellular communication and interactions with the extracellular environment. The heparan sulfate proteoglycans (HSPGs), including the transmembrane Syndecans, the glycosylphosphatidylinositol (GPI)-linked Glypicans and the secreted Agrin and Perlecan proteins (*Bernfield et al., 1999*; *Häcker et al., 2005*; *Kamimura and Maeda, 2021*; *Lin, 2004*; *Sarrazin et al., 2011*), play multiple roles in neuronal development. These include regulating neuronal migration and axon guidance, controlling diffusion of secreted signaling ligands, forming ECM barriers that maintain cell boundaries, and clustering transmembrane and secreted proteins (*Arikawa-Hirasawa et al., 2002*; *Cho et al., 2012*; *Fox and Zinn, 2005*; *Johnson et al., 2006*; *Kamimura et al., 2013*; *Kamimura and Maeda, 2021*; *Kinnunen, 2014*; *Nitkin et al., 1987*; *Sanes et al., 1978*). HSPGs encode core proteins with multiple extracellular motifs that are heavily modified by covalently attached heparan sulfate sugar chains that undergo enzymatic modifications (*Bishop et al., 2007*). Perlecan has the largest core mass of all HSPGs and is a conserved component of the extracellular matrix (ECM) and basement membranes (BMs) with Laminin, Nidogen and type IV Collagen (*Carson et al., 1993*; *Erickson and Couchman, 2000*; *Hassell et al., 1980*; *Martin et al., 1988*; *Martin and Timpl, 1987*; *Mouw et al., 2014*; *Noonan et al., 1991*). The ECM plays essential structural and signaling roles by maintaining tissue integrity and restricting diffusion of secreted signaling ligands (*Aviezer et al., 1994*; *Lindner et al., 2007*; *Park et al., 2003*; *Schaefer and Schaefer, 2010*). In this study, we identified a role for Perlecan in maintaining the stability of the ECM surrounding nerve bundles, with loss of the protein resulting in axonal breakage and degeneration, followed by synaptic retraction.

*Drosophila* larval motoneurons (MNs) and their glutamatergic neuromuscular junctions (NMJs) are a robust system for studying neuronal development and function due to abundant genetic toolkits and their ease of use for live and fixed imaging (*Andlauer and Sigrist, 2012*; *Bellen et al., 2019*; *Collins and DiAntonio, 2007*; *Harris and Littleton, 2015*; *Kanca et al., 2017*; *Owald and Sigrist, 2009*; *Sambashivan and Freeman, 2021*; *Şentürk and Bellen, 2018*). Many HSPGs are highly conserved in *Drosophila* and several function in neuronal development (*Dani et al., 2012*; *Han et al., 2020*; *Johnson et al., 2006*; *Kamimura et al., 2019*; *Kamimura et al., 2013*; *Koper et al., 2012*; *Nguyen et al., 2016*). The *Drosophila* Perlecan homolog is encoded by the gene *terribly reduced optic lobes* (*trol*) (*Datta and Kankel, 1992*; *Friedrich et al., 2000*; *Voigt et al., 2002*) and has been suggested to play a signaling role at NMJs by regulating Wingless (Wg) diffusion (*Kamimura et al., 2013*).

Given Perlecan has important structural functions as an ECM component in other developing tissues (*Costell et al., 1999*; *Skeath et al., 2017*), we examined if the protein played a similar role during synapse development or maintenance at *Drosophila* NMJs. Strikingly, *trol^null^* MNs developed progressive morphological defects over the course of larval development. Although NMJs developed normally in *trol^null^* larvae, they subsequently underwent retraction and displayed characteristic postsynaptic footprints where presynaptic material had been dismantled, similar to other *Drosophila* retraction mutants (*Eaton et al., 2002*; *Massaro et al., 2009*; *Pielage et al., 2011*; *Pielage et al., 2008*; *Pielage et al., 2005*). Although *trol^null^* MNs had normal synaptic output prior to retraction, MNs with disrupted NMJ structure lacked synaptic transmission. In addition, *trol^null^* MNs displayed an abnormal axonal cytoskeleton and underwent axonal breakage and loss. These phenotypes were independent of Perlecan's role in Wg diffusion and were not prevented by blocking Wallerian degeneration. Cell-type-specific knockdown and rescue experiments indicated *trol^null^* phenotypes were non-cell autonomous and required Perlecan secretion from multiple cell types. Within the peripheral nervous system (PNS), Perlecan was enriched in the neural lamella, a thick ECM structure that surrounds nerve bundles, the ventral nerve cord (VNC) and brain lobes (*Edwards et al., 1993*; *Stork et al., 2008*). Mutations in *trol* disrupted the neural lamella surrounding peripheral nerves, similar to previously identified defects in the CNS neural lamella (*Skeath et al., 2017*). Consistent with disruption of the neural lamella triggering axonal instability, loss of entire axonal bundles and NMJs temporally coincided within individual larval hemisegments. Together, these data indicate Perlecan plays a key role within the ECM to regulate the integrity and stability of MN axons and synapses.

## Results

### Perlecan is a conserved HSPG that localizes to the neural lamella surrounding peripheral nerves in *Drosophila* larvae

Perlecan is an evolutionary conserved HSPG with a similar domain architecture in invertebrates, vertebrates, and the early multicellular eukaryote *Trichoplax adhaerens* (*Warren et al., 2015*). *Drosophila* Perlecan is encoded by the *trol* locus, which resides on the X chromosome and encodes 25 predicted Perlecan splice variants ranging in size from 2853 to 4489 amino acids (*Figure 1A*). To compare the relationship of *Drosophila* Perlecan to other secreted HSPGs, a phylogenetic tree was constructed using homologs of Perlecan, Agrin and *Drosophila* Carrier of Wingless (Cow). One of the longest isoforms of *Drosophila* Perlecan (Trol-RAT) was used for the analysis. *Ciona intestinalis*, *Danio rerio*, *Mus musculus*, *Rattus norvegicus*, *Homo sapiens*, *Caenorhabditis elegans*, and *Trichoplax adhaerens* homologs were identified with NCBI blast searches. FASTA sequences of the longest isoform from each species was used in a Clustal Omega multiple sequence alignment and visualized in Jalview as an average distance phylogenetic tree using the BLOSUM62 algorithm (*Figure 1B*). The *Trichoplax* Perlecan and Agrin homologs were the most distantly related, but still clustered within their specific subfamily. *Drosophila* Perlecan clustered in a leaf with other Perlecan homologs and distinct from the Agrin family. Although Agrin plays a key role in cholinergic NMJ development (*Gautam et al., 1996*; *Nitkin et al., 1987*; *Sanes and Lichtman, 2001*), *Drosophila* contains glutamatergic NMJs and lacks an Agrin homolog (*Littleton and Ganetzky, 2000*). *Drosophila* Cow was more closely aligned with Agrin homologs than the Perlecan family.

Given the absence of an Agrin homolog in *Drosophila*, Perlecan might play similar roles in organizing *Drosophila* synaptic proteins. Indeed, a previous study identified defects in GluRIIA receptor clustering and synaptic Wg diffusion in *trol* mutants (*Kamimura et al., 2013*). These data suggested Perlecan may regulate organization of *Drosophila* synapses, prompting us to further evaluate its function. To examine Perlecan localization within the larval PNS, an endogenous *trol*^GFP^ insertion allele from the FlyTrap protein-tagging library was characterized (*Morin et al., 2001*). Trol^GFP^ was enriched along nerve bundles (*Figure 1C–F*) and present at lower levels on the surface of body wall muscles (*Figure 1C and G–I*). The enrichment of Perlecan around nerve bundles is consistent with its localization within the neural lamella, a large ECM compartment that surrounds axons and glia of the CNS and PNS. Indeed, Perlecan has been previously observed within the neural lamella surrounding the VNC and peripheral nerves (*Brink et al., 2012*; *Skeath et al., 2017*). In addition, the localization of Viking (Vkg), the *Drosophila* secreted type IV Collagen homolog and a known component of the neural lamella (*Yasothornsrikul et al., 1997*), was disrupted in *trol* mutants (see below). Although immunogold electron microscopy (EM) identified Perlecan in the subsynaptic reticulum (SSR) surrounding NMJs (*Kamimura et al., 2013*), Trol^GFP^ did not display synaptic enrichment beyond the homogenous expression over the entire muscle surface (*Figure 1G–I*). To confirm Trol^GFP^ signal was specific to Perlecan, an RNAi construct targeting *trol* (UAS-*trol*-RNAi.1, *Figure 1A*) was recombined with Trol^GFP^ and driven with the ubiquitous *tubulin*-Gal4 driver. Trol^GFP^ was eliminated by co-expression of the RNAi, with no signal observed along nerve bundles or on the muscle surface (*Figure 1D–I*). The Trol^GFP^ line also displayed normal NMJ growth and maintenance (*Figure 1—figure supplement 1*), in contrast to *trol* mutants (see below), indicating endogenous Trol^GFP^ produces a functional Perlecan protein.

### *trol*^null^ NMJs undergo synaptic retraction

To examine a role for Perlecan in synaptic development and function, a previously generated null mutant that deletes the *trol* locus (*Figure 1A*) was characterized (*Voigt et al., 2002*). Male *trol*^null^ larvae are smaller than their heterozygous female *trol*^null^/+counterparts and display disrupted locomotion and lethality during the 3rd instar stage (*Datta and Kankel, 1992*; *Voigt et al., 2002*). To examine synaptic morphology in *trol*^null^ and heterozygous control 3rd instars, immunostaining was performed at muscle 4 NMJs for presynaptic Complexin (Cpx) and postsynaptic Discs-large (Dlg). In *trol*^null^/+controls, Cpx and Dlg colocalized at NMJ boutons (*Figure 2A*). Although some *trol*^null^/y larvae had intact NMJs, many *trol* NMJs displayed Dlg +boutons that lacked presynaptic Cpx (*Figure 2A*). The presence of postsynaptic 'footprints' lacking presynaptic material is a defining feature of mutants that undergo synaptic retraction (*Eaton et al., 2002*; *Graf et al., 2011*; *Koch et al., 2008*; *Pielage et al., 2008*; *Pielage et al., 2005*). Trol^null^ mutants also displayed fewer synaptic boutons (*Figure 2B*),

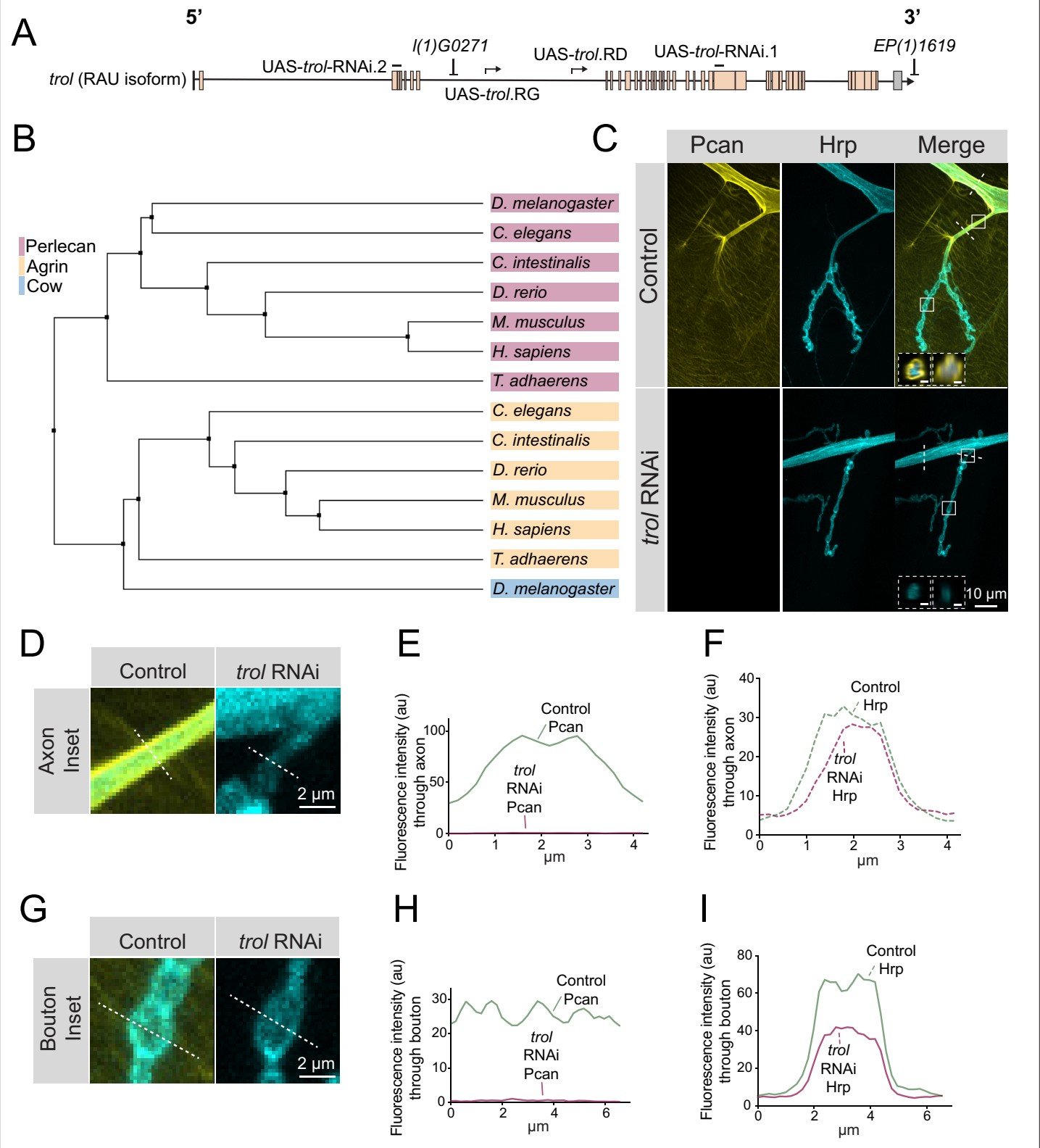

**Figure 1.** Perlecan conservation and localization within the *Drosophila* PNS. (**A**) Diagram of the *trol*-RAU isoform with exons (boxes) and introns (lines) indicated. Sequence locations targeted by two UAS-*trol* RNAi lines and start sites for two overexpression constructs (UAS-*trol*.RD and UAS-*trol*.RG) used in this study are noted. The location of P-elements *l(1)G0271* and *EP(1)1619* previously mobilized to generate the *trol^null^* deletion allele (**Voigt et al., 2002**) is also shown. (**B**) Phylogenetic tree of Perlecan, Agrin and Carrier of Wingless (Cow) from the indicated species generated using BLOSUM62

*Figure 1 continued on next page*

*Figure 1 continued*

average distance. (**C**) Representative images of muscle 4 NMJs stained for Perlecan (Pcan) and Hrp in control (*trol^GFP^, UAS-trol-RNAi.1/+;+;+*) or Perlecan knockdown (*trol^GFP^, UAS-trol-RNAi.1/+;+;tub-Gal4/+*) larvae. White boxes denote location of insets depicted in D and G. Dashed white lines denote location of insets depicted in lower left corner. Left inset displays orthogonal section through axon bundles, showing Perlecan signal in the neural lamella (scale bar 2 μm). Right inset displays orthogonal section through individual axon, showing Perlecan signal in the neural lamella (scale bar 1 μm). (**D, G**) Magnified view of control and *trol* RNAi axons (**D**) and boutons (**G**), highlighting loss of Perlecan following RNAi knockdown. Dashed lines show representative sites for line scanning quantification for panels E-F, H-I. (**E–F, H–I**) Line scan profiles of Perlecan (**E, H**) or Hrp (**F, I**) mean fluorescent intensity through axons (**E, F**) or synaptic boutons (**H, I**) at muscle 4 in segment A2 (control: 14 axons from 8 larvae; *trol* RNAi: 19 axons from 11 larvae; control: 14 NMJs from 8 larvae; *trol* RNAi: 22 NMJs from 11 larvae). Control measurements are denoted in green and *trol* RNAi in magenta.

The online version of this article includes the following source data and figure supplement(s) for figure 1:

**Source data 1.** Raw Values and Statistics for *Figure 1* Perlecan Localization.

**Figure supplement 1.** Endogenous Trol^GFP^ strain produces a functional Perlecan protein.

**Figure supplement 1—source data 1.** Raw Values and Statistics for *Figure 1—figure supplement 1* on Bouton Number.

in addition to increased synaptic footprints (*Figure 2C*), compared to control NMJs in abdominal segments A3-A5 (bouton number p values: 0.0379 for A3, 0.0012 for A4,<0.0001 for A5). Consistent with other retraction mutants (*Graf et al., 2011*), the *trol^null^* phenotype was more severe in posterior abdominal segments (*Figure 2B and C*). NMJs showing severe synaptic retraction, defined by retraction footprints and decreased number of synaptic boutons two standard deviations compared to controls, were only observed in *trol^null^* larvae, with increasing severity in posterior abdominal segments (*Figure 2D*). Together, these data suggest NMJs are lost over development in larvae lacking Perlecan.

To determine if synaptic loss is specific to the absence of Perlecan and not the *trol^null^* genetic background, NMJs were examined in *trol^null^* larvae in trans to a deficiency (*Df(1)ED411*) that removes the *trol* locus. *Df(1)ED411/+* heterozygous NMJs appeared normal and lacked postsynaptic footprints. In contrast, *Df(1)ED411/trol^null^* NMJs had significantly reduced bouton number (p=0.0048), with 50% of NMJs showing postsynaptic footprints (*Figure 2—figure supplement 1A–C*). In addition, expression of UAS-*trol*-RNAi.2 with the ubiquitous *tubulin*-Gal4 driver resulted in reduced bouton number, with >65% of NMJs showing postsynaptic footprints compared to RNAi or Gal4 only controls (*Figure 2—figure supplement 1D–F*).

The experiments described above examined synaptic retraction of type Ib glutamatergic MNs. However, larval muscles are also innervated by type Is glutamatergic and type II and III neuromodulatory MNs. The two glutamatergic MNs have distinct morphology and physiology, with tonic-like (Ib) or phasic-like (Is) properties (*Aponte-Santiago and Littleton, 2020*). To assay if Perlecan is required for stability of other MN subtypes, synaptic retraction was quantified at Is NMJs in *trol^null^* larvae. Similar to Ib, *trol^null^* Is NMJs displayed a significant reduction in bouton number (p=0.0011 A3, p=0.0238 A4, p=0.0068 A5), with >30% showing severe retraction phenotypes compared to heterozygous controls (*trol^null^/+*, *Figure 2—figure supplement 2A–C*). In addition to glutamatergic MNs, type II and III neuromodulatory MNs also displayed missing NMJs (data not shown). Together, these data indicate synaptic retraction occurs across all MN subtypes in 3rd instar larvae lacking Perlecan.

## Synaptic retraction is independent of Perlecan's role in Wingless signaling

A previous study identified changes in pre- and postsynaptic Wg levels at *trol* mutant NMJs, suggesting Perlecan restricts Wg diffusion within the synaptic cleft. The absence of Perlecan resulted in enhanced presynaptic and reduced postsynaptic Wg signaling (*Kamimura et al., 2013*). Although synaptic retraction has not been previously associated with the Wg pathway at *Drosophila* NMJs (*Franco et al., 2004*; *Mathew et al., 2005*; *Mosca and Schwarz, 2010*; *Packard et al., 2002*; *Restrepo et al., 2022*), genetic interaction studies were conducted to assay whether increased presynaptic Wg output might contribute to synaptic retraction phenotypes observed in *trol^null^* mutants. A UAS construct expressing a constitutively active (CA) form of the *Drosophila* GSK3 serine/threonine kinase Shaggy (Sgg, UAS-*sgg^S9A^*), which dominantly blocks Wg signaling (*Cook et al., 1996*; *Siegfried et al., 1992*), was expressed in glutamatergic MNs of *trol^null^* larvae using *vGlut*-Gal4. NMJ bouton number and synaptic retraction were quantified at muscles 6 and 7 in UAS-*sgg^S9A^* controls, in *trol^null^*, *vGlut*-Gal4, and in *trol^null^* mutants expressing CA-Sgg (*trol^null^; vGlut*-Gal4 >*sggS^9A^*). Like muscle 4, muscle

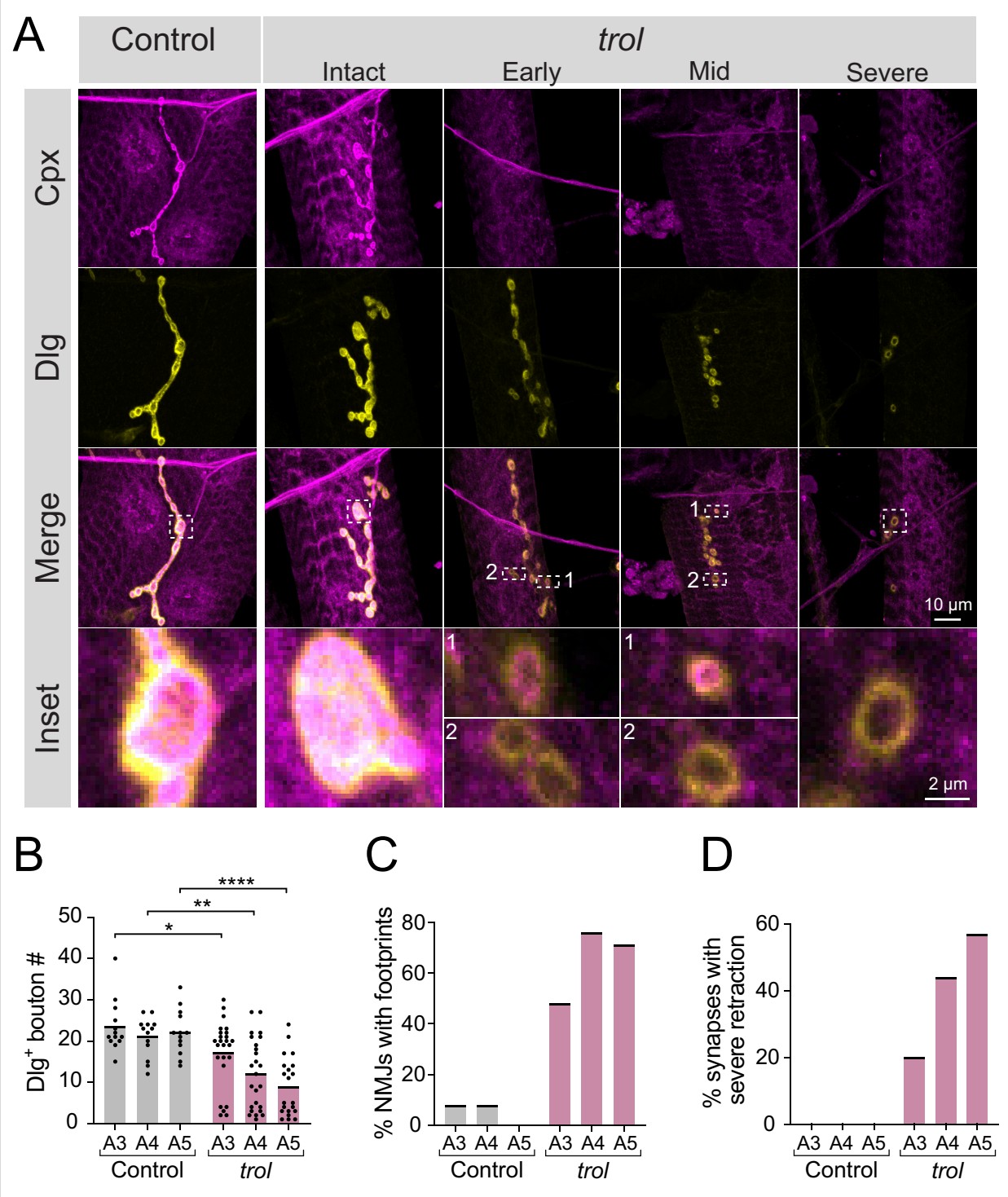

**Figure 2.** Synaptic retraction in *trol^null* motoneurons. (**A**) Representative images of muscle 4 NMJs in control (*trol^null*/+;+;+, left panel) and *trol^null* (*trol^null*/y;+;+) larvae stained for Cpx (magenta) and Dlg (yellow). Example *trol* NMJs of increasing retraction severity are shown in the right panels. Highlighted areas in the merged image are shown as insets at the bottom. For mild and moderately retracted NMJs, inset 1 displays a bouton with pre- and postsynaptic material still co-localized while inset 2 highlights a synaptic footprint with only postsynaptic material remaining. (**B**) Quantification of Dlg +Ib bouton number from control and *trol* muscle 4 NMJs for abdominal segments A3-A5. Each point represents the number of boutons at one NMJ, with mean bouton number indicated with the solid black line. Quantification of bouton number: control A3: 23.5±1.8, 13 NMJs from 7 larvae; *trol* A3: 17.2±1.6, 25 NMJs from 13 larvae, p<0.05; control A4: 21.1±1.3, 13 NMJs from 7 larvae; *trol* A4: 12.0±1.7, 25 NMJs from 13 larvae, p<0.01; control A5: 22.1±1.5, 13 NMJs from 7 larvae; *trol* A5: 8.9±1.6, 21 NMJs from 13 larvae, p<0.0001. (**C**) Percentage of control or *trol* NMJs with one or more

*Figure 2 continued on next page*

Figure 2 continued

postsynaptic footprints (Dlg bouton lacking Cpx) for segments A3-A5 from the dataset in B. (**D**) Percentage of control or *trol* NMJs with severe retraction for segments A3-A5 from the dataset in B.

The online version of this article includes the following source data and figure supplement(s) for figure 2:

**Source data 1.** Raw Values and Statistics for *Figure 2* on Synapse Number and Retraction.

**Figure supplement 1.** Synapse retraction occurs in *trol^null^* mutants over deficiency and in *trol* RNAi knockdown larvae.

**Figure supplement 1—source data 1.** Raw Values and Statistics for *Figure 2—figure supplement 1* on Trol Mutant and RNAi Retraction.

**Figure supplement 2.** *trol^null^* Is synapses retract.

**Figure supplement 2—source data 1.** Raw Values and Statistics for *Figure 2—figure supplement 2* on Segmental Retraction.

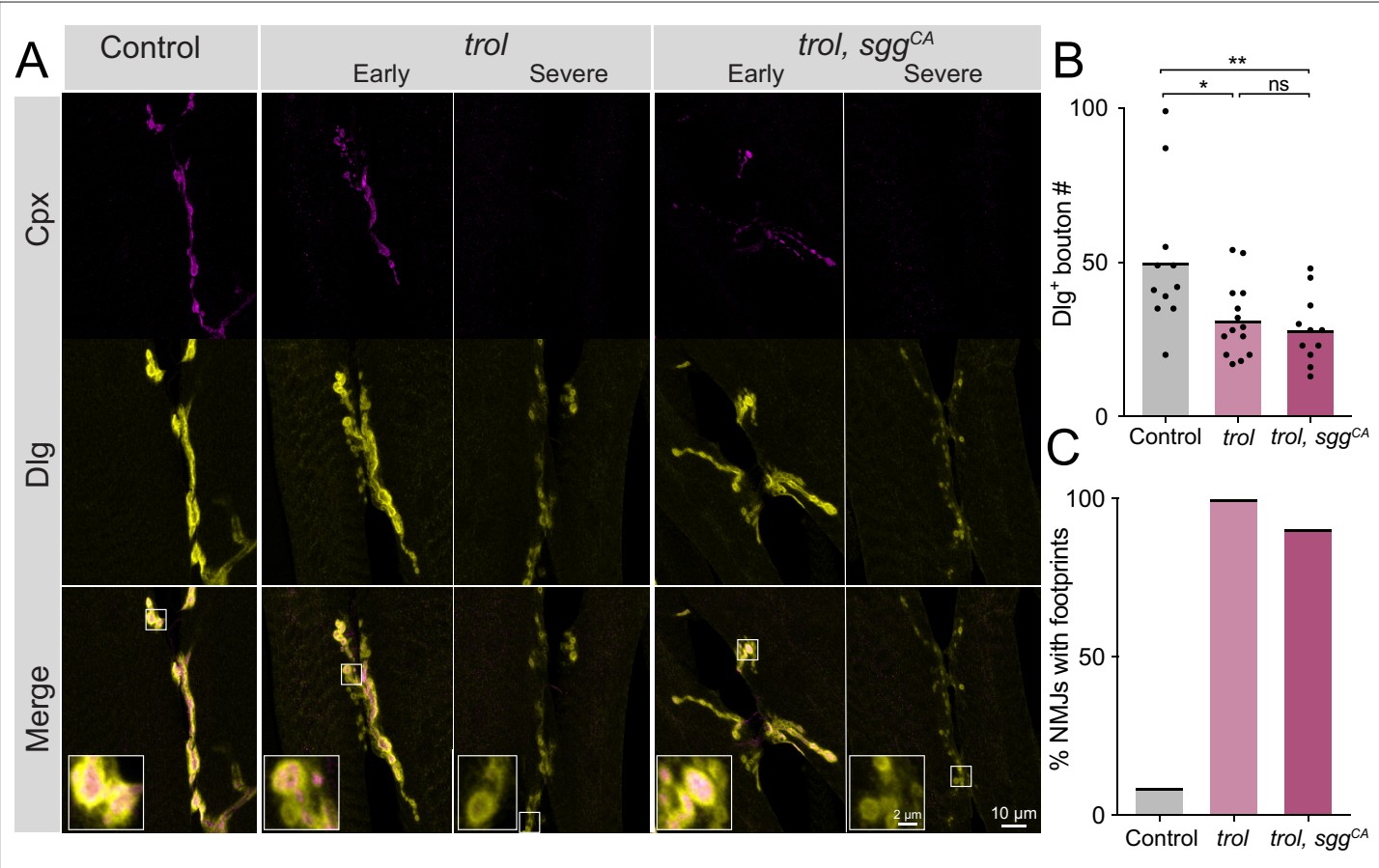

**Figure 3.** Synaptic retraction in *trol* mutants is not prevented by blocking presynaptic Wg signaling. (**A**) Representative images of muscle 6/7 NMJs at segment A4 in control (*+;UAS-sgg^S9A^/+;+*, left panel), *trol^null^* (*trol^null^,vGlut-Gal4/y;+;+*, middle panels) and *trol^null^, sgg^CA^* (*trol^null^,vGlut-Gal4/y;UAS-sgg^S9A^/+;+*, right panels) larvae stained for Cpx (magenta) and Dlg (yellow). Example NMJs that are in early or severe stages of retraction are shown for both *trol* genotypes. Highlighted areas in the merge are shown as insets in the bottom panel with either intact (early) or retracting (severe) boutons. (**B**) Quantification of Dlg +Ib and Is bouton number from control, *trol^null^* and *trol^null^, sgg^CA^* muscle 6/7 NMJs for abdominal segment A4. Each point represents the number of boutons at one NMJ, with mean bouton number indicated with the solid black line. Quantification of bouton number: control: 50.1±7.0, 11 NMJs from 6 larvae; *trol*: 31.3±3.2, 14 NMJs from 7 larvae, p<0.05 compared to control; *trol, sgg^CA^*: 28.2±3.4, 11 NMJs from 6 larvae; p<0.01 compared to control, p=0.8814 compared to *trol*. (**C**) Percentage of control, *trol*, or *trol, sgg^CA^* NMJs with one or more postsynaptic footprints for the dataset in B.

The online version of this article includes the following source data for figure 3:

**Source data 1.** Raw Values and Statistics for *Figure 3* on Perlecan - Wingless Pathway Interactions.

6/7 NMJs showed reduced bouton number (*P*=0.0174) and postsynaptic footprints in >90% of *trol*$^{null}$ larvae, indicating synaptic retraction is not restricted to MNs innervating muscle 4 (*Figure 3A–C*). Blocking presynaptic Wg signaling with CA-Sgg did not suppress the loss of boutons or prevent synaptic retraction in *trol*$^{null}$ mutants (*Figure 3A–C*, p=0.8814 between *trol* and *trol, sgg*$^{CA}$ bouton numbers). These data indicate Perlecan's function in controlling synaptic stability is independent of its effects on Wg signaling.

## *trol*$^{null}$ NMJs develop normally and retract during the 3rd instar larval stage

Although postsynaptic footprints are a hallmark of synaptic retraction, it is possible that Perlecan loss disrupts early MN or synaptic development such that synapses classified as retracted never contained presynaptic material. To dynamically visualize NMJ development, serial intravital imaging of larval muscle 26 NMJs was performed over 4 days in *trol*$^{null}$ and heterozygous controls following brief anesthesia as previously described (*Akbergenova et al., 2018*). Larvae containing endogenously tagged nSynaptobrevin (nSyb$^{GFP}$) (*Guan et al., 2020*) and a construct expressing GluRIIA$^{RFP}$ (*Schmid et al., 2008*) were used to visualize presynaptic vesicles and postsynaptic receptors, respectively (*Figure 4A*). Although control and *trol*$^{null}$ larvae had similar NMJ area at the 2nd instar stage when imaging began (*Figure 4B*), only control NMJs continued to grow on subsequent days of imaging (*Figure 4C*). In contrast, both pre- and postsynaptic area declined in *trol*$^{null}$ larvae during the 3rd instar imaging window. Control NMJs had greater pre- than postsynaptic area, while *trol*$^{null}$ larva had smaller nSyb$^{GFP}$ than GluRIIA$^{RFP}$ area by the final day of imaging, consistent with loss of presynaptic material and lingering postsynaptic footprints (*Figure 4C*). Several patterns of presynaptic loss were observed in *trol*$^{null}$ NMJs during serial imaging (*Figure 4A*). In some cases, an entire branch of an axonal arbor was lost between imaging days. At other NMJs, presynaptic material was absent from internal boutons in an axon branch, with proximal and distal boutons from the same axon containing nSyb$^{GFP}$. These findings confirm that *trol*$^{null}$ MNs form NMJs that are subsequently retracted during development.

In vivo imaging indicated *trol*$^{null}$ NMJs are morphologically intact prior to retraction. However, loss of Perlecan could cause functional disruption of synaptic output earlier in development. To assay synaptic function at intact versus retracting NMJs in *trol*$^{null}$ 3$^{rd}$ instars, two-electrode voltage-clamp (TEVC) electrophysiology was performed at muscle 6. Following physiological recordings, dissected *trol*$^{null}$ larvae were bathed with fluorescent anti-Hrp to visualize NMJs and determine if they were intact or retracted at the recording site (*Figure 4D*). At intact *trol*$^{null}$ NMJs, nerve stimulation resulted in evoked release amplitude similar to controls, indicating normal presynaptic output prior to retraction (*Figure 4E and F*, p=0.3189). In contrast, fully retracted *trol*$^{null}$ NMJs completely lacked evoked release (*Figure 4E and F*, p=0.0001). Quantal imaging with postsynaptic membrane-tethered GCaMP7s (*Akbergenova et al., 2018*; *Melom et al., 2013*) revealed a similar loss of spontaneous release in retracted NMJs (data not shown). When evoked responses from intact and retracted *trol*$^{null}$ NMJs were combined and averaged, a significant reduction in evoked excitatory junctional current (eEJC) was observed compared to controls (*Figure 4F*, p=0.027). The gradual loss of NMJs is consistent with other cell types where Perlecan is dispensable for the initial formation of BMs and only required for their maintenance (*Costell et al., 1999*; *Matsubayashi et al., 2017*).

## Non-cell autonomous Perlecan secretion is required for synaptic stability

The largest source of Perlecan in *Drosophila* comes from the larval fat body, where it is secreted into the hemolymph and incorporated into the ECM surrounding most cell types (*Pastor-Pareja and Xu, 2011*). Although abundant in the fat body, *trol* mRNA is also present at lower levels in larval MNs and muscles (*Jetti et al., 2023*). To determine the cell type(s) responsible for Perlecan secretion that controls synaptic stability, UAS-*trol* RNAi.2 was driven with a panel of cell-type-specific Gal4 drivers. As indicated above, ubiquitous knockdown of *trol* mRNA with *tubulin*-Gal4 abolishes Perlecan expression (*Figure 1D–G*) and causes synaptic retraction (*Figure 2—figure supplement 1*). In contrast, knockdown of Perlecan with UAS-*trol*-RNAi.2 (*Figure 1A*) driven by multiple Gal4 lines expressed in specific cell populations, including pan-neuronal (*elav*$^{C155}$), neuronal and muscle (*elav*$^{C155}$ and *mef2*-Gal4), glial (*repo*-Gal4), fat body (*ppl*-Gal4, *Lsp2*-Gal4), hemocytes (*Hml*-Gal4), and fat body and hemocytes

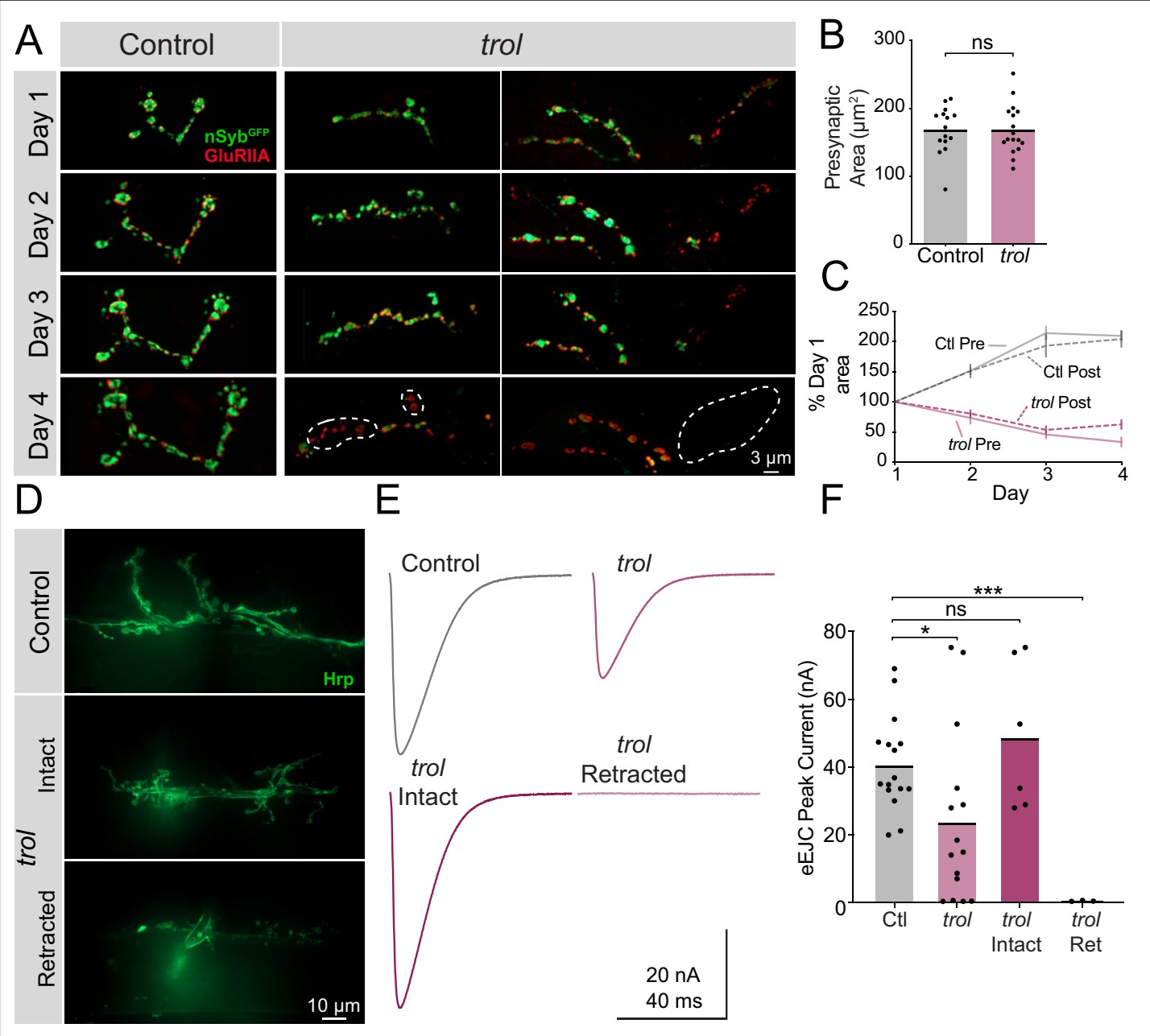

**Figure 4.** *Trol* mutant NMJs form normally but are not maintained over development. (**A**) Representative NMJ images during serial intravital imaging of muscle 26 in control (*trol^null^/+;+;nSyb^GFP^/GluRIIA^RFP^*, left) and *trol^null^* (*trol^null^/y;+;nSyb^GFP^/GluRIIA^RFP^*, right) larvae over 4 days starting at the 2nd instar stage. nSyb^GFP^ is shown in green and GluRIIA^RFP^ in red. Dashed lines highlight areas where presynaptic nSyb is missing while postsynaptic GluRIIA remains (retraction footprints). Severe NMJ retraction is seen in both example *trol^null^* NMJs by day 4, while control NMJs continue to grow. (**B**) Quantification of muscle 26 presynaptic NMJ area in control and *trol^null^* 2nd instar larvae at the beginning of serial intravital imaging sessions. No differences in NMJ area are present during this stage of early development. Quantification of NMJ area: control: 168.5±8.974, 15 NMJs from 4 larvae; *trol*: 168.5±8.855, 17 NMJs from 4 larvae, p=0.9991. NMJs from multiple abdominal segments were imaged. (**C**) Percent change in NMJ area during each day of imaging for control and *trol^null^* larvae. Both presynaptic and postsynaptic area continue to increase in controls, while synaptic area is lost in *trol^null^* larvae. (**D**) Representative images of muscle 6/7 NMJs stained with anti-Hrp following TEVC physiology in control (*trol^null^/+;+;+*) and *trol^null^* (*trol^null^/y;+;+*) 3rd instar larvae. Examples of intact (middle) and retracted (bottom) NMJs are shown. (**E**) Average eEJC traces in 0.3 mM Ca²⁺ saline in control and all *trol^null^* NMJs combined (top), together with average traces from NMJs of only intact or retracted *trol^null^* NMJs (bottom). (**F**) Quantification of average eEJC peak amplitude (nA) per NMJ in segments A3 and A4 for the indicated genotypes. Intact and retracted *trol* NMJs were determined post-hoc blinded following anti-Hrp staining and paired with their corresponding eEJC data. Quantification of EJC amplitude: control: 40.8±3.5 nA, 16 NMJs from 7 larvae; *trol* combined: 23.8±6.6, 15 NMJs from 6 larvae, p<0.05 compared to control; *trol* intact: 48.8±8.9 nA, 6 NMJs from 4 larvae, p=0.3189 compared to control; *trol* retracted: 0.5±0.1, 3 NMJs from 2 larvae, p<0.0001 compared to control.

*Figure 4 continued on next page*

Figure 4 continued

The online version of this article includes the following source data for figure 4:

**Source data 1.** Raw Values and Statistics for *Figure 4* on Trol Serial Imaging and Physiology.

(*c564*-Gal4), failed to trigger synaptic retraction or reduce synaptic bouton number (*Figure 5A and B*). In addition, knockdown of Perlecan in the three cell types that form or surround NMJs (MNs, muscles and glia) failed to reduce Trol$^{GFP}$ signal around larval nerves or on the muscle surface (*Figure 5C–G*). Although it is possible that these individual drivers are weaker than *tubulin*-Gal4, no phenotypes were observed when *trol*-RNAi was driven by *elav$^{C155}$*, *elav$^{C155}$* and *mef2*, and *repo* at 29 °C to increase Gal4 activity (data not shown). Together, these data suggest Perlecan secretion from multiple cell types is required to stabilize NMJs.

The majority of proteins that control synaptic stability at *Drosophila* NMJs function cell autonomously within the neuron (*Eaton et al., 2002*; *Graf et al., 2011*; *Koch et al., 2008*; *Pielage et al., 2008*; *Pielage et al., 2005*). To determine whether MN secretion of Perlecan is sufficient to stabilize synapses, UAS-*trol* constructs encoding two different Perlecan isoforms (*Figure 1A*) were overexpressed with *vGlut*-Gal4 in the *trol$^{null}$* background. Overexpression of Perlecan specifically in MNs did not rescue the reduction in bouton number or synaptic retraction phenotypes in *trol$^{null}$* larvae (*Figure 5—figure supplement 1A–C*, p=0.54 for bouton number with *trol$^{RG}$* rescue), suggesting neuronally secreted Perlecan is insufficient for maintaining synaptic stability. Together, these data indicate Perlecan acts non-cell autonomously from multiple cell types to stabilize larval NMJs.

## NMJ loss in *trol* mutants is not exacerbated by mechanical stress from enhanced muscle contraction

Studies of Perlecan's role within the ECM of other *Drosophila* cell types and in several mammalian tissues indicate the protein helps withstand mechanical stress during tissue development (*Arikawa-Hirasawa et al., 2002*; *Costell et al., 1999*; *Pastor-Pareja and Xu, 2011*; *Skeath et al., 2017*; *Töpfer et al., 2022*). Although Perlecan is not enriched at synaptic boutons, *trol$^{null}$* NMJs could retract due to a failure to withstand mechanical stress from repeated contractions during larval crawling that would normally be buffered by the small amount of Perlecan normally on muscles. To test this model, a mutation in Myosin heavy chain (*Mhc$^{S1}$*) that causes a dominant hypercontractive muscle phenotype (*Montana and Littleton, 2006*; *Montana and Littleton, 2004*) was brought into the *trol$^{null}$* background to assay if synaptic retraction phenotypes were enhanced. Despite increased muscle contraction in *trol$^{null}$; Mhc$^{S1}$/+* larvae, no enhancement of synaptic retraction or decreases in bouton number were observed (*Figure 6A and B*, p=0.5975). Given hypercontraction in *Mhc$^{S1}$* mutants requires synaptic transmission (*Montana and Littleton, 2004*) and retracted *trol$^{null}$* NMJs lack evoked release (*Figure 4C–E*), we cannot exclude the possibility that *Mhc$^{S1}$* mutants only enhance early stages of synaptic retraction prior to loss of presynaptic output. To examine if muscle hypercontraction increases Perlecan NMJ abundance as a protective mechanism to withstand elevated muscle contraction force, endogenous Trol$^{GFP}$ was brought into the M*hc$^{S1}$* mutant background. No enhancement of Perlecan staining was observed around axons, at NMJs, or on muscles in Trol$^{GFP}$; *Mhc$^{S1}$/+* larvae compared to Trol$^{GFP}$ alone (*Figure 6C–E*). Together with the lack of Perlecan enrichment around boutons, the failure of muscle hypercontraction to increase instability of *trol$^{null}$* NMJs suggest the protein is unlikely to play a mechanical role within the extracellular space around boutons to directly stabilize NMJs.

## The absence of Perlecan disrupts the neural lamella and triggers coordinated synaptic loss across abdominal hemisegments

*Drosophila* MN cell bodies reside within the VNC and their axons exit in segmental nerve bundles that also contain incoming sensory neuron axons. Nerve bundles are wrapped by several layers of glial cells and surrounded by the neural lamella (*Edwards et al., 1993*; *Stork et al., 2008*), a specialized ECM structure containing Perlecan. Although *trol$^{null}$* mutants display synaptic retraction phenotypes at larval NMJs, Perlecan is primarily expressed within the neural lamella surrounding larval nerves and not at NMJs (*Figure 1D–I*). As such, where Perlecan acts to regulate synaptic stability is unclear. If loss of Perlecan compromises the function of the neural lamella as a physical and protective barrier for encapsulated axons over time, NMJ retraction might occur for all axons within a nerve bundle in

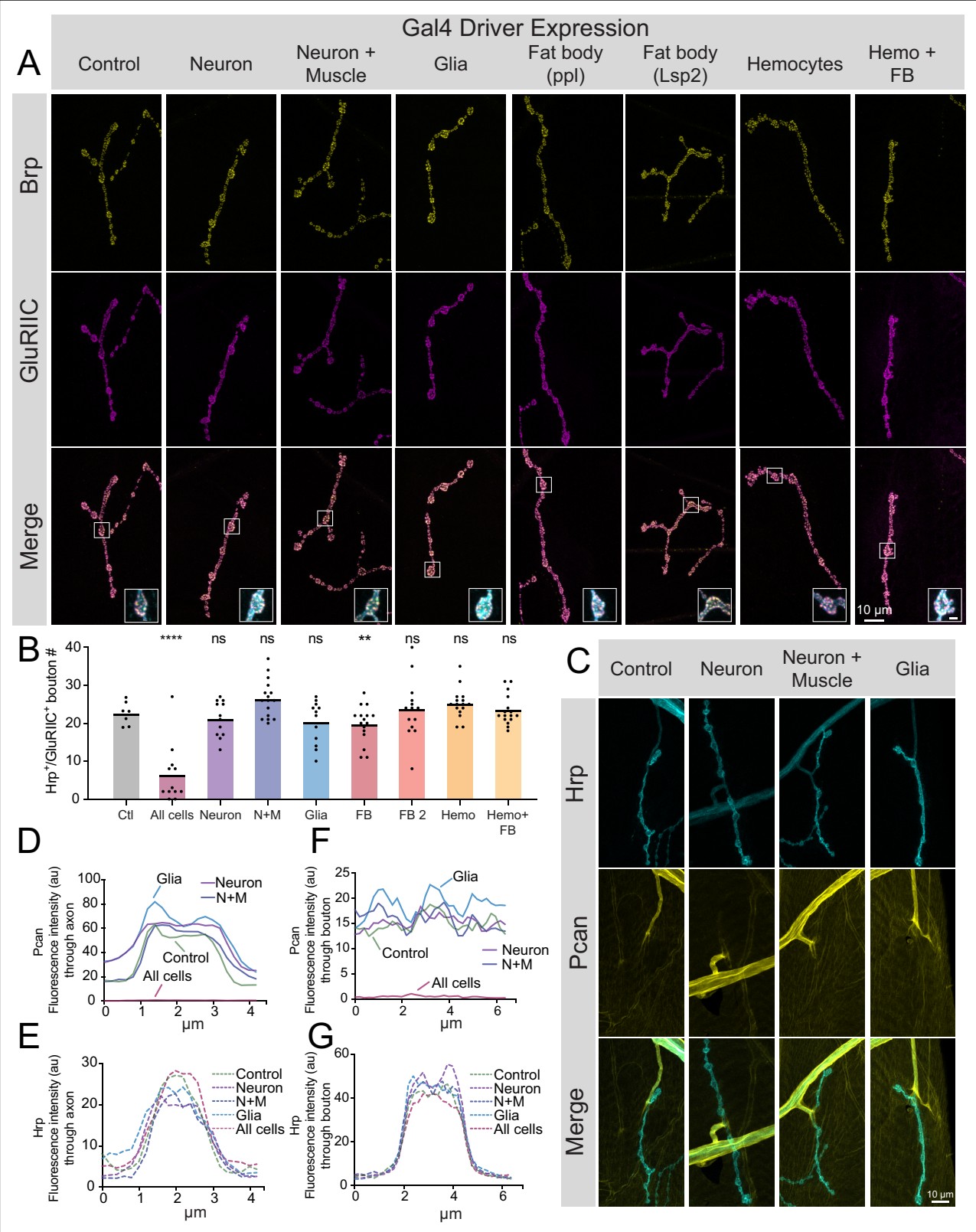

**Figure 5.** Perlecan acts in non-cell autonomous fashion to control synaptic maintenance. (**A**) Representative images of larval muscle 4 NMJs at segment A4 stained for Brp (yellow), GluRIIC (magenta) and anti-Hrp (cyan, only shown in inset) in control (*+;UAS-trol-RNAi.2/+;+*) and *trol* RNAi knockdown in the indicated cell types (UAS-*trol*-RNAi.2 (one copy) driven by *elav^C155^*, *elav^C155^* and *mef2*-Gal4, *repo*-Gal4, *ppl*-Gal4, *Lsp2*-Gal4, *Hml*-Gal4, and *c564*-Gal4 (one copy)). The merged image is shown below with location of the insets highlighting single boutons. Inset scale bar is 2 µm. (**B**) Quantification

*Figure 5 continued on next page*

*Figure 5 continued*

of Hrp+/GluRIIC +positive Ib bouton number at muscle 4 NMJs in segment A4 in controls and following cell-type specific *trol* RNAi knockdown. Each knockdown was analyzed in separate experiments with both UAS only and Gal4 only controls. Significance was calculated for each experimental comparison, but a single control that represents the average bouton number of every experiment is plotted for ease of visualization. Quantification of *trol* knockdown with *tubulin*-Gal4 at A4 (from *Figure 2—figure supplement 1*) is included for comparison. Unlike pan-cellular RNAi, cell-type specific RNAi does not induce synaptic retraction. Quantification of bouton number: neuron UAS only control: 23.25±1.399, 12 NMJs from 6 larvae; neuron Gal4 only control: 23.75±1.226, 12 NMJs from 6 larvae; *elav*$^{C155}$ >UAS-*trol*-RNAi.2: 21.17±1.359, 12 NMJs from 6 larvae; p=0.4424 compared to UAS control; p=0.2994 compared to Gal4 control; neuron and muscle UAS only control: 21.21±1.130, 14 NMJs from 7 larvae; neuron and muscle Gal4 only control: 23.43±1.189, 14 NMJs from 7 larvae; *elav*$^{C155}$, *mef2* >UAS-*trol*-RNAi.2: 26.44±1.248, 16 NMJs from 8 larvae; p=0.0066 (<0.01) compared to UAS control; p=0.1437 compared to Gal4 control; glia UAS only control: 26.83±1.375, 12 NMJs from 6 larvae; glia Gal4 only control: 23.79±1.407, 14 NMJs from 7 larvae; *repo* >UAS-*trol*-RNAi.2: 20.25±1.643, 12 NMJs from 6 larvae; p=0.0078 (<0.01) compared to UAS control; p=0.1666 compared to Gal4 control; fat body (*ppl*) UAS only control: 25.57±1.312, 14 NMJs from 7 larvae; fat body (*ppl*) Gal4 only control: 26.21±0.7644, 14 NMJs from 7 larvae; *ppl* >UAS-*trol*-RNAi.2: 19.75±1.216, 16 NMJs from 8 larvae; p<0.01 compared to UAS and Gal4 controls (0.0014 compared to UAS; 0.0004 compared to Gal4); fat body 2 (*Lsp2*) UAS only control: 19.08±1.412, 13 NMJs from 7 larvae; fat body 2 (*Lsp2*) Gal4 only control: 25.07±0.7593, 14 NMJs from 7 larvae; *Lsp2* >UAS-*trol*-RNAi.2: 23.64±2.053, 14 NMJs from 7 larvae; p=0.0731 compared to UAS control; p=0.7249 compared to Gal4 control; hemocyte UAS only control: 18.93±0.8285, 14 NMJs from 7 larvae; hemocyte Gal4 only control: 26±1.441, 14 NMJs from 7 larvae; *Hml* >UAS-*trol*-RNAi.2: 25.19±0.9841, 16 NMJs from 8 larvae; p=0.0005 (<0.001) compared to UAS control; p=0.8242 compared to Gal4 control; hemocyte and fat body UAS only control: 23.14±1.181, 14 NMJs from 7 larvae; hemocyte and fat body Gal4 only control: 24.25±0.8972, 12 NMJs from 6 larvae; *c564* >UAS-*trol*-RNAi.2: 23.44±0.9999, 16 NMJs from 8 larvae; p=0.9705 compared to UAS control; p=0.8149 compared to Gal4 control. (**C**) Representative images of muscle 4 NMJs stained for Perlecan and Hrp in control (Trol$^{GFP}$,UAS-*trol*-RNAi.1) or *trol* RNAi knockdown by *elav*$^{C155}$, *elav*$^{C155}$ and *mef2*-Gal4, or *repo*-Gal4 (one copy of UAS and Gal4 constructs). (**D–G**) Line scanning profiles of Perlecan (**D,F**) or Hrp (**E,G**) mean fluorescent intensity through axons (**D,E**) or synaptic boutons (**F,G**) at muscle 4 in segment A4. Measurements are color-coded as indicated: control (green), *trol* RNAi in neurons (magenta), neurons and muscles (blue), or glia (light blue). No reduction in Perlecan around nerves or on the muscle surface surrounding boutons was observed compared with *tubulin*-Gal4 knockdown (*Figure 1F–I*, replicated here ('All cells') for comparison).

The online version of this article includes the following source data and figure supplement(s) for figure 5:

**Source data 1.** Raw Values and Statistics for *Figure 5* on Cell-type Specfic Trol Knockdown.

**Figure supplement 1.** Overexpression of Perlecan in *trol*$^{null}$ motoneurons does not rescue synaptic retraction phenotypes.

**Figure supplement 1—source data 1.** Raw values and statistics for *Figure 5—figure supplement 1* on cell-type specific trol rescue.

a temporally coordinated manner across larval hemisegments. To examine if synapses within each segmental nerve bundle showed evidence of coordinated loss, abdominal bodywall hemisegments of *trol*$^{null}$ larvae expressing *vGlut*-Gal4; UAS-*10xGFP* were examined. NMJ area on muscle 6/7, 4, and 1 was quantified for each hemisegment and compared to controls. NMJs were often completely absent in one hemisegment, while fully intact in others. Indeed, *Trol*$^{null}$ NMJs within each individual hemisegment displayed similar decreases in synaptic area (*Figure 7A–C*), indicating entire hemisegments undergo coordinated synaptic loss while others remain intact.

To determine if the neural lamella surrounding peripheral nerves was disrupted in the absence of Perlecan, the expression and localization of the neural lamella-localized Collagen IV homolog Vkg was assayed. In control segmental nerves, Vkg showed a similar localization to Perlecan and surrounded axon bundles exiting the VNC and at nerve branch points to muscles (*Figure 8A and E*). In contrast, Vkg expression in *trol*$^{null}$ nerve bundles was dimmer with gaps in staining around axonal segments, along with abnormal aggregation at specific sites along the nerve (*Figure 8A, B and E*). To quantify neural lamella disruption, fluorescence intensity of Vkg and anti-Hrp (to label axons) was determined for segmental nerve bundles. Unlike controls, Vkg staining in axonal cross sections from *trol*$^{null}$ larvae showed a thinner, or in some cases absent, neural lamella surrounding Hrp +axon bundles (*Figure 8B*). Anti-Hrp staining was also brighter in *trol*$^{null}$ axon bundles, suggesting there may be greater antibody penetration in the absence of a functional neural lamella (*Figure 8A and B*). The mean fluorescence of Vkg and Hrp signal was calculated in axon bundles traveling over muscle 4. Vkg signal around axons was significantly reduced in *trol*$^{null}$ larvae, with the Vkg/Hrp ratio significantly lower in *trol*$^{null}$ axons than controls (*Figure 8C and D*, p=0.0014 and 0.0004). This phenotype was independent of whether NMJs had undergone retraction, as nerves for both intact and retracted NMJs displayed a reduced neural lamella. Compared to controls, Vkg staining was also non-evenly distributed along nerves, with multiple sites showing extracellular accumulation of Vkg beyond the traditional boundaries of the neural lamella (*Figure 8E*). In some protrusions, Hrp +axonal material protruded from its normal boundary to co-localize with Vkg. As such, Perlecan may play a role in capturing or retaining Vkg within the neural lamella surrounding larval nerves.

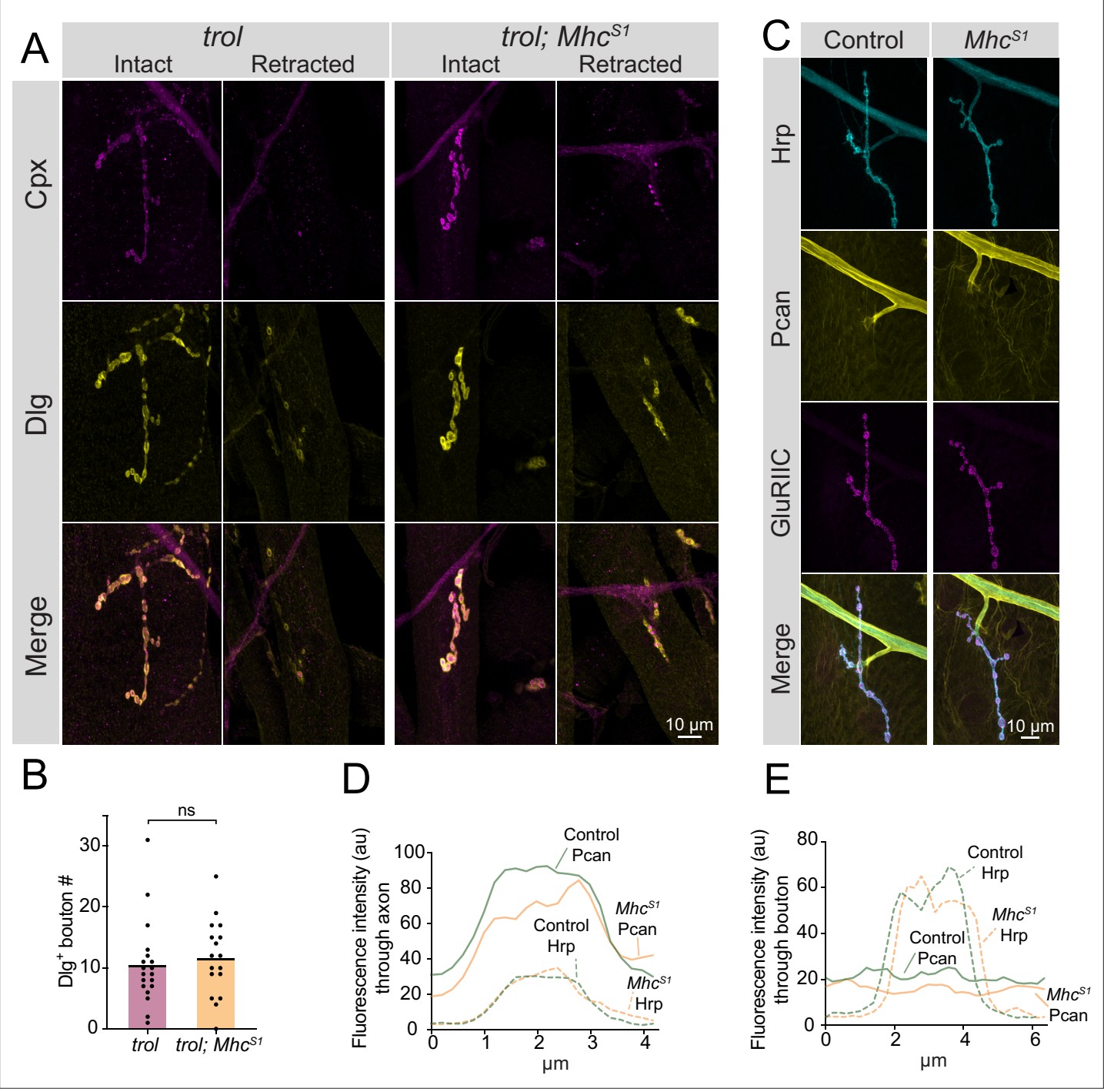

**Figure 6.** Enhanced muscle contraction does not exacerbate synaptic retraction in *trol* mutants. (**A**) Representative images of muscle 4 NMJs at segment A3 in *trol* (*trol^null^/y;+;+*) and *trol; Mhc^S1^* (*trol^null^/y;Mhc^S1^/+;+*) larvae stained for Cpx (magenta) and Dlg (yellow). Examples of intact and retracted NMJs are shown for both genotypes. Brightness for images of retracted NMJs was enhanced to show residual synaptic material. (**B**) Quantification of Dlg +Ib bouton number from *trol* and *trol; Mhc^S1^* muscle 4 NMJs at segment A3. Each point represents the number of boutons at one NMJ, with mean bouton number indicated with the solid black line. Quantification of bouton number: *trol*: 10.5±1.6, 19 NMJs from 10 larvae; *trol; Mhc^S1^*: 11.7±1.5, 19 NMJs from 10 larvae, p=0.597. (**C**) Representative images of muscle 4 NMJs stained for Perlecan, Hrp and GluRIIC in control (*trol^GFP^/y;+;+*) or *Mhc^S1^* (*trol^GFP^/y;Mhc^S1^/+;+.*) larvae. (**D,E**) Line scanning profiles of Perlecan and Hrp fluorescent intensity through axons (**D**) or synaptic boutons (**E**) at muscle 4 in segment A4. For both axons and boutons, line profiles from 12 control NMJs from 6 larvae and 10 *Mhc^S1^* NMJs from 5 larvae were averaged. Control measurements are denoted in green and *Mhc^S1^* in orange.

The online version of this article includes the following source data for figure 6:

**Source data 1.** Raw Values and Statistics for *Figure 6* on Trol - Mhc Interactions.

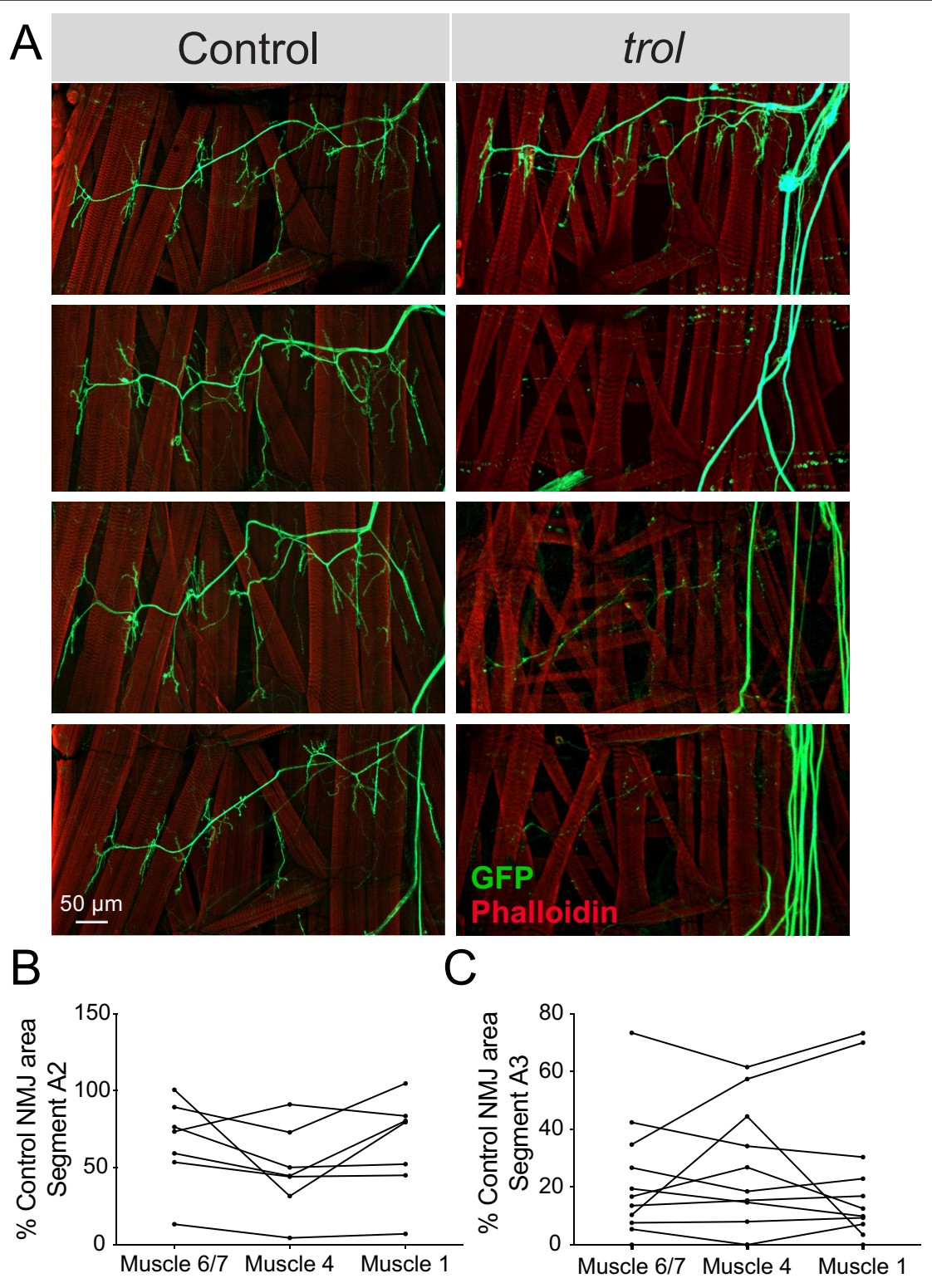

**Figure 7.** NMJ loss occurs in a temporally coordinated manner across abdominal hemisegments in *trol* mutants. (**A**) Representative images of larval hemisegments in four control (*trol^null^,vGlut-Gal4/+;+;UAS-10xGFP/+*, left panels) or *trol^null^* (*trol^null^,vGlut-Gal4/y;+;UAS-10xGFP/+*, right panels) larvae expressing 10X-GFP in motoneurons (green) and stained for Phalloidin to label muscle Actin (red). (**B**) Area of listed NMJs (muscle 6/7, muscle 4, muscle 1) in *trol* segment A2 as a percentage of mean control NMJ area. Percent area is largely consistent across NMJs along the hemisegment, indicating

*Figure 7 continued on next page*

Figure 7 continued

synapses retract or are maintained together with their hemisegment. (**C**) Area of listed NMJs (muscle 6/7, muscle 4, muscle 1) in *trol* segment A3 as a percentage of mean control NMJ area.

The online version of this article includes the following source data for figure 7:

**Source data 1.** Raw Values and Statistics for *Figure 7* on Hemi-segment Coordinated NMJ Retraction.

## Mutations in *trol* cause axonal damage independent of the Wallerian degeneration pathway

Prior studies indicated disruptions to the microtubule cytoskeleton often proceed NMJ loss in other synaptic retraction mutants (*Eaton et al., 2002*; *Pielage et al., 2005*). Given loss of Perlecan compromises the neural lamella, we examined axonal morphology in *trol^null* mutants by visualizing the axonal and synaptic microtubule network with immunostaining for Futsch, the *Drosophila* homolog of microtubule associated protein 1B (MAP1B) (*Hummel et al., 2000*). Microtubule bundles in MN axons innervating muscle 4 were examined in larvae expressing *vGlut*-Gal4; UAS-*10xGFP* or UAS-*myrRFP*. Microtubules in heterozygous *trol^null/+* control larvae formed filamentous tracks within axons that extended into synaptic boutons (*Figure 9A*). In contrast, *trol^null* axons contained fragmented and noncontinuous microtubule tracks or lacked Futsch staining altogether at branch points where the axon exited towards the muscle (*Figure 9A*). Quantification of Futsch staining intensity in *trol^null* axons revealed a significant reduction (*Figure 9B*, p<0.0001). Similar defects were observed within synaptic terminals, where NMJs undergoing retraction lacked Futsch staining or displayed fragmented microtubules (*Figure 9A*). These data indicate disruptions to the microtubule cytoskeleton within *trol^null* axons and NMJs accompany synaptic retraction.

To determine if axons showed more severe defects in morphology, axon bundles and microtubules were imaged at different developmental timepoints. In mature 3rd instar controls, axon bundles have defined boundaries with smooth tracks of Futsch + microtubules (*Figure 9C*). In contrast, *trol^null* axon bundles displayed progressive defects throughout larval development. 2nd instar *trol^null* larvae had some axonal swellings, small RFP + protrusions beyond the normal nerve boundary, and slightly twisted and disorganized Futsch + microtubule tracts (*Figure 9C*). By early 3rd instar, nerve bundles were disorganized, with numerous protrusions and tangled microtubules (*Figure 9C*). At the mature 3rd instar stage, some axon bundles were entirely severed, with large tangled nets of disorganized Futsch at the ends of severed nerves (*Figure 9C–D*), similar to previously described retraction balls that form after axonal injury in mammals (*Cajal, 1928*). When nerve bundles in a hemisegment were severed, NMJs in that hemisegment displayed severe retraction. Single motoneuron labeling with a MN1-Ib Gal4 driver (*Aponte-Santiago et al., 2020*) expressing UAS-CD4-TdT to label axons in *trol^null* larvae revealed discontinuous axon membrane labeling (*Figure 9E*). The time course of these deficits suggests that axonal damage and breakage occurs upstream of synapse loss, given that 2nd instar *trol^null* larvae have normal synaptic area but display altered axonal structure (*Figure 9C–D*, *Figure 3B*).

Loss of distal axons following damage is a well-known trigger for neurodegeneration in both invertebrates and vertebrates (*Gerdts et al., 2016*; *Perlson et al., 2010*). Neurite loss, synaptic retraction and eventual neuronal death following axonal damage often proceeds through a defined molecular cascade known as Wallerian degeneration (*Coleman and Höke, 2020*; *Conforti et al., 2014*; *Llobet Rosell and Neukomm, 2019*; *Luo and O'Leary, 2005*; *Sambashivan and Freeman, 2021*; *Wang et al., 2012*). Inhibiting this protein cascade promotes distal axon survival following injury in multiple systems (*DiAntonio, 2019*; *Fang et al., 2012*; *Figley et al., 2021*; *Gerdts et al., 2015*; *Gilley et al., 2017*; *Gilley et al., 2015*; *Llobet Rosell et al., 2022*; *Neukomm et al., 2017*; *Sasaki et al., 2016*). To determine whether NMJ loss in *trol^null* mutants utilizes the Wallerian degeneration signaling cascade, an established RNAi inhibitor of an upstream component of the pathway, dSarm (*Gerdts et al., 2015*; *Gerdts et al., 2013*; *Osterloh et al., 2012*), was expressed in *trol^null* MNs. Inhibition of this pathway did not rescue distal axon maintenance, as *trol^null* larvae expressing *dSarm* RNAi still had reduced synaptic bouton number (*P*=0.5387 between *trol* and Wallerian degeneration inhibition conditions), with >80% of NMJs displaying postsynaptic footprints associated with synaptic retraction (*Figure 9F–H*). These data indicate *Drosophila* MNs undergo axonal degeneration and synaptic retraction in *trol^null* mutants independent of the Wallerian degeneration pathway. Together, we conclude that loss of Perlecan disrupts the neural lamella, leading to axonal damage that causes cytoskeletal disruption and synaptic

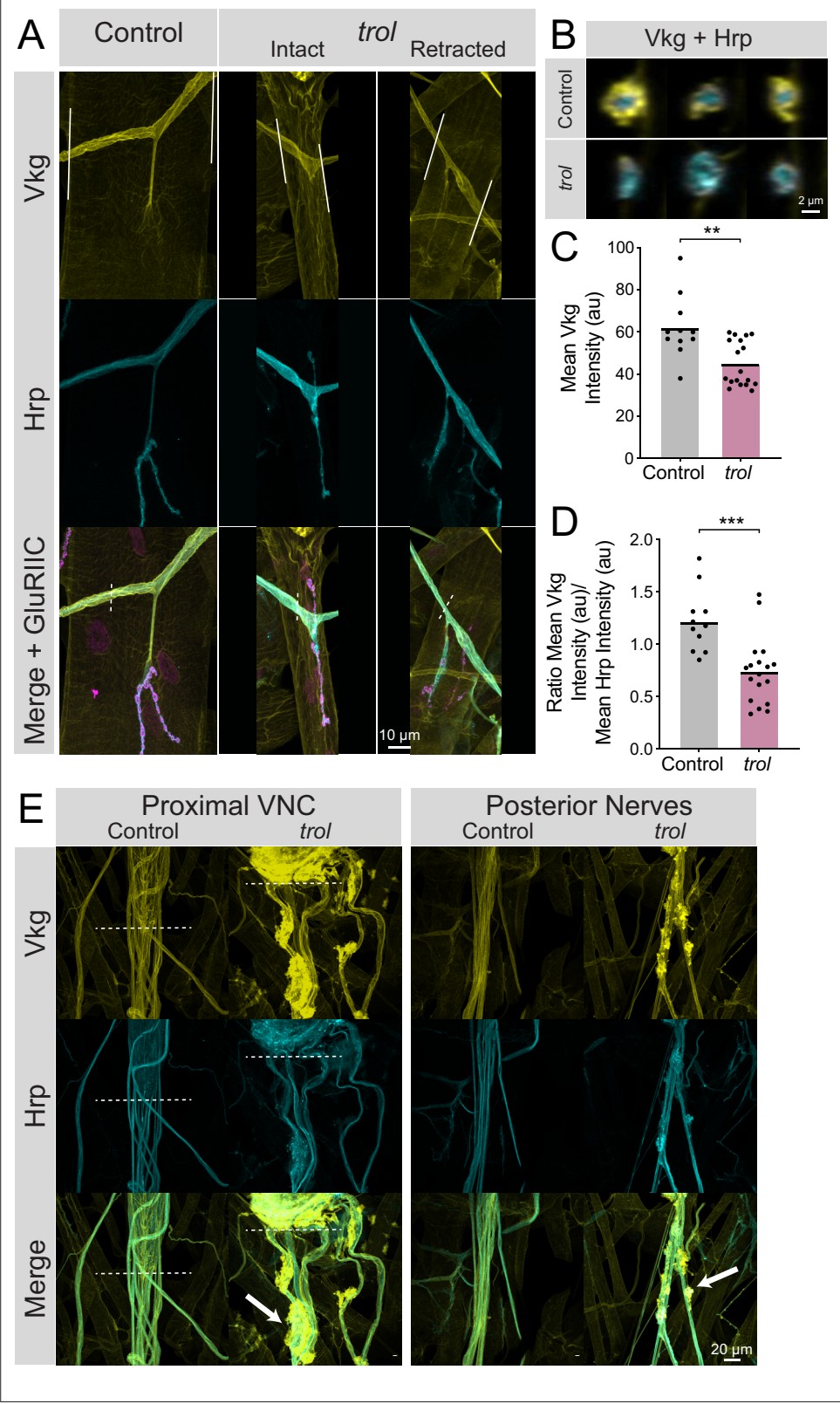

**Figure 8.** Loss of Perlecan disrupts the neural lamella. (**A**) Representative images of muscle 4 NMJs in control (*trol^null^/+;vkg^GFP^/+;+*) and *trol^null^* (*trol^null^/y;vkg^GFP^/+;+*) larvae stained for Vkg (yellow), Hrp (cyan), and GluRIIC (magenta). Images of intact and retracted *trol* NMJs are shown. White lines in Vkg panels indicate borders for quantification of axon bundle fluorescence in panels C and D. (**B**) Representative cross-sections of control (top)

*Figure 8 continued on next page*

*Figure 8 continued*

and *trol* (bottom) axon bundles with Vkg in yellow and Hrp in cyan. (**C**) Quantification of mean Vkg fluorescence in the axon bundle crossing over muscle 4. Each point represents one axonal segment measurement. (Control: 7 larvae; n=11; 62.00±4.476; *trol*: 9 larvae; n=18; 45.04±2.549; *P*=0.0014). (**D**) Quantification of the ratio of mean Vkg fluorescence divided by mean Hrp fluorescence in the axon bundle crossing over muscle 4. Each point represents the ratio for one axonal segment measurement. (Control: 7 larvae; n=11; 1.219±0.08998; *trol*: 9 larvae; n=18; 0.7402±0.07467, p<0.001). (**E**) Representative images of axon bundles stained for Vkg (yellow) and Hrp (cyan) exiting the proximal VNC or those located more posteriorly. Control nerve bundles are on the left, with *trol* nerve bundles on the right. White dashed lines indicate the posterior tip of the VNC. White arrows note areas of Vkg accumulation and protrusions from the neural lamella.

The online version of this article includes the following source data for figure 8:

**Source data 1.** Raw Values and Statistics for *Figure 8* on Vkg Alterations in Neural Lamella.

retraction in a temporally coordinated manner across individual hemisegment nerve bundles during larval development (*Figure 10*).

## Discussion

In this study, we identified a role for the ECM protein Perlecan in regulating the structure and integrity of the neural lamella surrounding segmental nerve bundles in *Drosophila* larvae. Loss of Perlecan caused defects in neural lamella ECM function, with reduced thickness of the lamella based on staining for the type IV collagen Vkg. In addition, Vkg accumulated at aberrant sites along nerve bundles, with neuronal axons present outside of their normal boundary and within neural lamella protrusions. Although MNs formed functional NMJs in the absence of Perlecan, these synapses destabilized and rapidly retracted during later stages of larval development. Defects in axonal morphology and disruptions to the microtubule cytoskeleton were present before NMJs retracted, suggesting insults to axonal integrity and function were early events triggering synaptic retraction in *trol* mutants.

The normal development of *Drosophila* larval MNs and NMJs in *trol* mutants, followed by destabilization and subsequent breakdown, is consistent with a late role for Perlecan in ECM function and stability described in other systems (*Hayes et al., 2022*). In vertebrates, Perlecan loss causes degeneration of the developing heart only after pumping begins (*Costell et al., 1999*). The *C. elegans* Perlecan homolog Unc-52 regulates a late stage of muscle-epidermis attachment (*Rogalski et al., 2001*; *Rogalski et al., 1995*) and can promote ectopic presynaptic growth after synapse formation when other ECM components are missing (*Qin et al., 2014*). Within developing *Drosophila* egg chambers, Perlecan and type IV Collagen function to establish mechanical properties of the ECM, protecting the egg from osmotic stress (*Töpfer et al., 2022*). In some contexts, Perlecan and type IV Collagen have opposing roles in regulating ECM rigidity (*Pastor-Pareja and Xu, 2011*; *Skeath et al., 2017*). Within the neural lamella surrounding the *Drosophila* VNC and brain, Perlecan acts to reduce ECM stiffness established by Vkg and β-integrin (*Skeath et al., 2017*). EM imaging of the VNC neural lamella in *trol* mutants demonstrates a much thinner ECM (*Skeath et al., 2017*), consistent with reduced Vkg thickness around larval nerve bundles identified in this study. Studies in *Drosophila* embryos indicate Perlecan is a late delivered component of BMs and ECMs, requiring type IV Collagen for its incorporation into the matrix (*Matsubayashi et al., 2017*; *Pastor-Pareja and Xu, 2011*). Together, these data support a key role for Perlecan as a regulator of tissue maturation and maintenance.

Perlecan can be produced in numerous cell types in *Drosophila*, with specific roles requiring secretion from neurons (*Cho et al., 2012*), muscles (*Kamimura et al., 2013*), glia (*Skeath et al., 2017*), hemocytes and fat body (*Isabella and Horne-Badovinac, 2015*; *Pastor-Pareja and Xu, 2011*; *Ramos-Lewis et al., 2018*), intestinal stem cells (*You et al., 2014*) and wing imaginal discs (*Bonche et al., 2021*). In contrast to these cases of a single cell type supplying the source of Perlecan, the neural lamella's role in regulating MN axonal and synaptic stability requires Perlecan to act in a non-cell autonomous role and be secreted from multiple cell types. RNAi knockdown of *trol* in neurons, muscles, glia, fat body, or hemocytes was not sufficient to induce synaptic retraction, in contrast to pan-cellular knockdown with *tubulin*-Gal4. In addition, expression of Perlecan specifically in neurons in *trol* mutants was not sufficient to rescue synaptic retraction phenotypes, in contrast to other characterized retraction mutants where the affected gene acts cell-autonomously (*Eaton et al., 2002*;

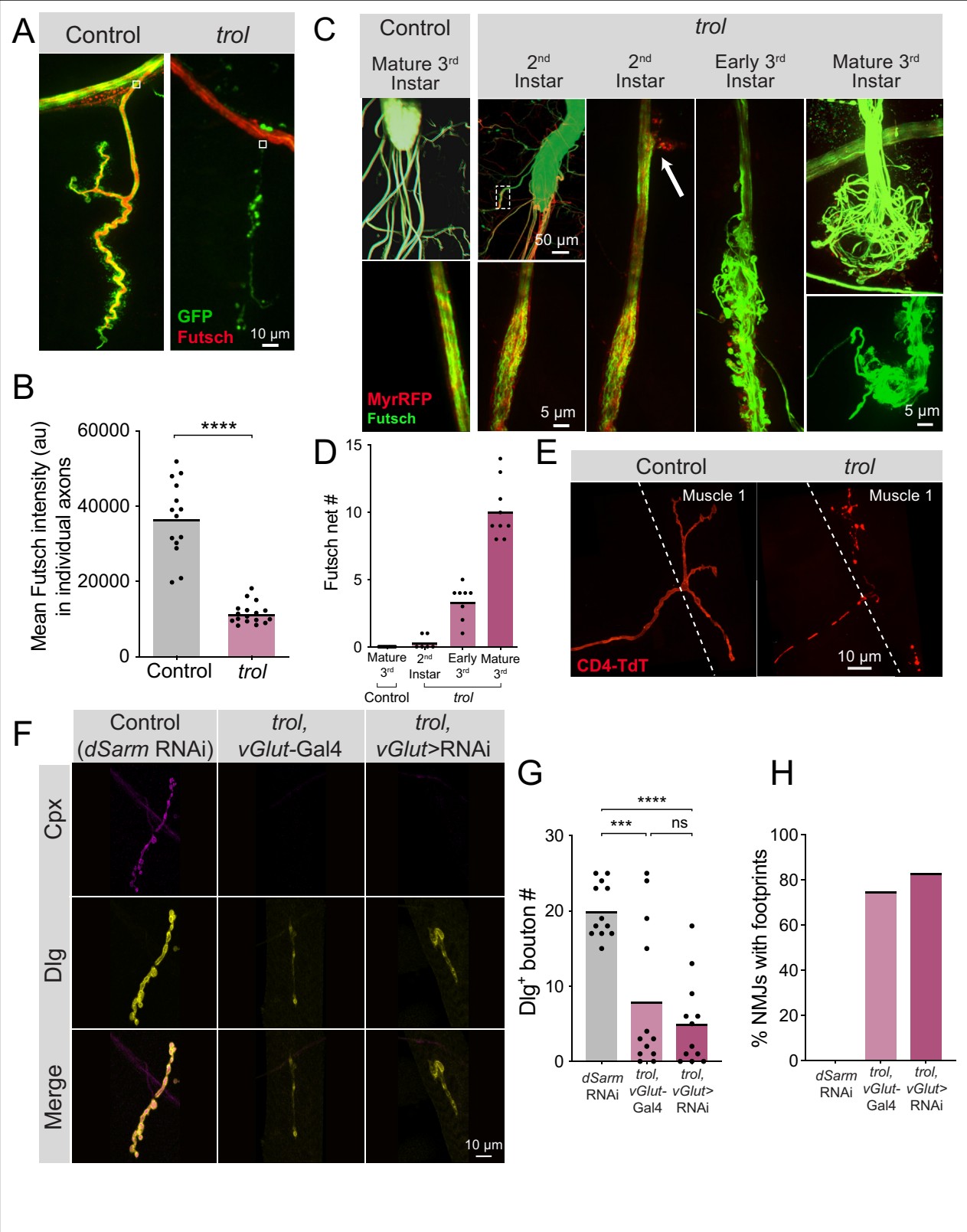

**Figure 9.** Fragmentation of the microtubule cytoskeleton and axons in *trol* mutants. (**A**) Representative images of Ib motoneurons innervating muscle 4 in control (*trol^null^,vGlut-Gal4/+;+;UAS-10xGFP/+*, left panel) or *trol* (*trol^null^,vGlut-Gal4/y;+;UAS-10xGFP/+,* right panel) larvae expressing 10X-GFP in motoneurons (green) and stained for Futsch to label microtubules (red). White boxes indicate area of axon with Futsch fluorescence quantified in B. (**B**) Quantification of mean Futsch fluorescence in individual axons. Each point represents the average fluorescence in one axon. Quantification of

*Figure 9 continued on next page*

*Figure 9 continued*

mean fluorescence: control: 36725±2679, 14 NMJs from 6 larvae; trol: 11476±675.7, 17 NMJs from 7 larvae, p<0.0001. Multiple abdominal segments were imaged. (**C**) Representative images of VNCs and axon bundles stained for Futsch (green) and expressing MyrRFP (red) in control (*trol^null^,vGlut-Gal4/+;UAS-myrRFP/+;+*) and *trol* (*trol^null^,vGlut-Gal4/y; UAS-myrRFP/+;+*) larvae at the indicated stage. Mature 3^rd^ instar control axon bundles have continuous tracks of microtubules within intact axons. *trol* axons show progressive disruption, with early swelling of the axon and mildly twisted microtubules (white dashed box depicts location of swollen axon and twisted microtubules shown in inset). The 2^nd^ instar axon bundle image is replicated in the next panel to indicate an area of membrane material leaving the axon boundary (white arrow). By late 3^rd^ instar, *trol* axons are severed and nets of tangled microtubules form balls at either end of the axon bundle breakage. (**D**) Quantification of the number of Futsch nets in mature 3^rd^ instar control and *trol* larvae at multiple developmental stages. Each point indicates the number of Futsch nets in one larvae. (**E**) Representative images of MN1-Ib-Gal4 driving expression of UAS-CD4-TdTomato (red) to visualize single axons and synapses in control (*trol^null^/+;+;MN1-Ib-Gal4,UAS-CD4-TdTomato*) and *trol* (*trol^null^/y;+;MN1-Ib-Gal4,UAS-CD4-TdTomato*) 3^rd^ instar larvae. (**F**) Representative images of larval muscle 4 NMJs stained for Cpx (magenta) and Dlg (yellow) in control (*+;UAS-dSarm-RNAi/+;+*), *trol, vGlut*-Gal4 (*trol^null^,vGlut-Gal4/y;+;+*) and *trol^null^* expressing *dSarm* RNAi (*trol^null^,vGlut-Gal4/y;UAS-dSarm-RNAi/+,+*). Brightness for images of retracted NMJs was enhanced to show residual synaptic material. (**G**) Quantification of Dlg +Ib bouton number at muscle 4 NMJs in segment 4 in control, *trol, vGlut*-Gal4, and *trol^null^* expressing *dSarm*-RNAi. Each point represents bouton number from one NMJ with the mean indicated by the solid black line. Quantificatíon of bouton number: control: 20.08±1.048, 12 NMJs from 6 larvae; *trol, vGlut*-Gal4: 8.083±2.811, 12 NMJs from 7 larvae, p<0.001 compared to control; *trol^null^* expressing *dSarm*-RNAi: 5.083±1.667, 12 NMJs from 6 larvae, p<0.0001 compared to control, P=0.5387 compared to *trol^null^*. (**H**) Percentage of control, *trol, vGlut*-Gal4 or *trol^null^* expressing *dSarm*-RNAi NMJs with footprints from the dataset in F.

The online version of this article includes the following source data for figure 9:

**Source data 1.** Raw Values and Statistics for *Figure 9* on Microtubule Disruption in Trol Mutants.

*Eaton and Davis, 2005*; *Graf et al., 2011*; *Koch et al., 2008*; *Massaro et al., 2009*; *Pielage et al., 2011*; *Pielage et al., 2008*; *Pielage et al., 2005*). Our data suggest multiple cell types are required to secrete Perlecan for proper incorporation and function within the neural lamella, consistent with examples where multiple cell types are required to secrete Perlecan for its functional role in several contexts (*Bonche et al., 2022*; *Bonche et al., 2021*; *You et al., 2014*).

Following our initial observations of synaptic retraction in *trol* mutants, we hypothesized that mechanical stress caused by repeated muscle contraction during larval crawling might destabilize synaptic boutons due to defects in synaptic cleft rigidity. Indeed, Perlecan can act to resist mechanical stress during tissue development in *Drosophila* and other species by providing malleability to the ECM and BM (*Costell et al., 1999*; *Farach-Carson et al., 2014*; *Khalilgharibi and Mao, 2021*; *Pastor-Pareja and Xu, 2011*; *Qin et al., 2014*; *Ramos-Lewis et al., 2018*; *Skeath et al., 2017*; *Töpfer et al., 2022*). However, increasing the mechanical stress of muscle contraction with a previously generated hyperactive *Mhc* mutant in the lab (*Montana and Littleton, 2004*) did not enhance synaptic retraction in *trol* mutants. We also found no role for local regulation of Wg diffusion with the synaptic cleft, as blocking presynaptic Wg output did not prevent retraction. The same genetic approach to block presynaptic Wg signaling prevented formation of some satellite boutons observed early on in *trol* mutants (*Kamimura et al., 2013*). As such, the low levels of Perlecan at the NMJ do not appear to be the site of action for how it normally prevents synaptic retraction.

Together with the lack of Perlecan enrichment at synaptic boutons, we focused on the neural lamella surrounding nerve bundles as a potential site of action in regulating synaptic stability. MN axons are under constant tension during larval crawling with fixed points of attachment at somas within the VNC and terminal anchors at the NMJ (*Fan et al., 2019*; *Fan et al., 2017*; *Siechen et al., 2009*; *Tofangchi et al., 2016*). Indeed, larval axons degenerated over development in *trol* mutants. Synaptic microtubules were absent in retracting NMJs, and fragmented microtubules with breaks at the site where axons entered the synaptic field were often observed, along with breakage of entire axon bundles upstream of NMJs. Imaging of the type IV collagen Vkg, an essential component of ECMs and the neural lamella, revealed reduced Vkg thickness and abnormal accumulation of the protein around nerve bundles in *trol* mutants. Axon retraction events were also temporally correlated in individual hemisegments, suggesting catastrophic breakdown of the neural lamella in single segmental nerves is likely the trigger for axon bundle breakage and loss of the entire nerve bundle.

In summary, this study indicates a critical role for Perlecan in neural lamella integrity that prevents axonal degeneration and synaptic retraction in *Drosophila* MNs. Despite the involvement of axons in Perlecan-dependent synaptic retraction, inhibition of Wallerian degeneration did not prevent synapse loss in *trol^null^* mutants. Although the morphology of glia involved in wrapping axons and forming the blood brain barrier was not characterized, it would not be surprising if they displayed morphological

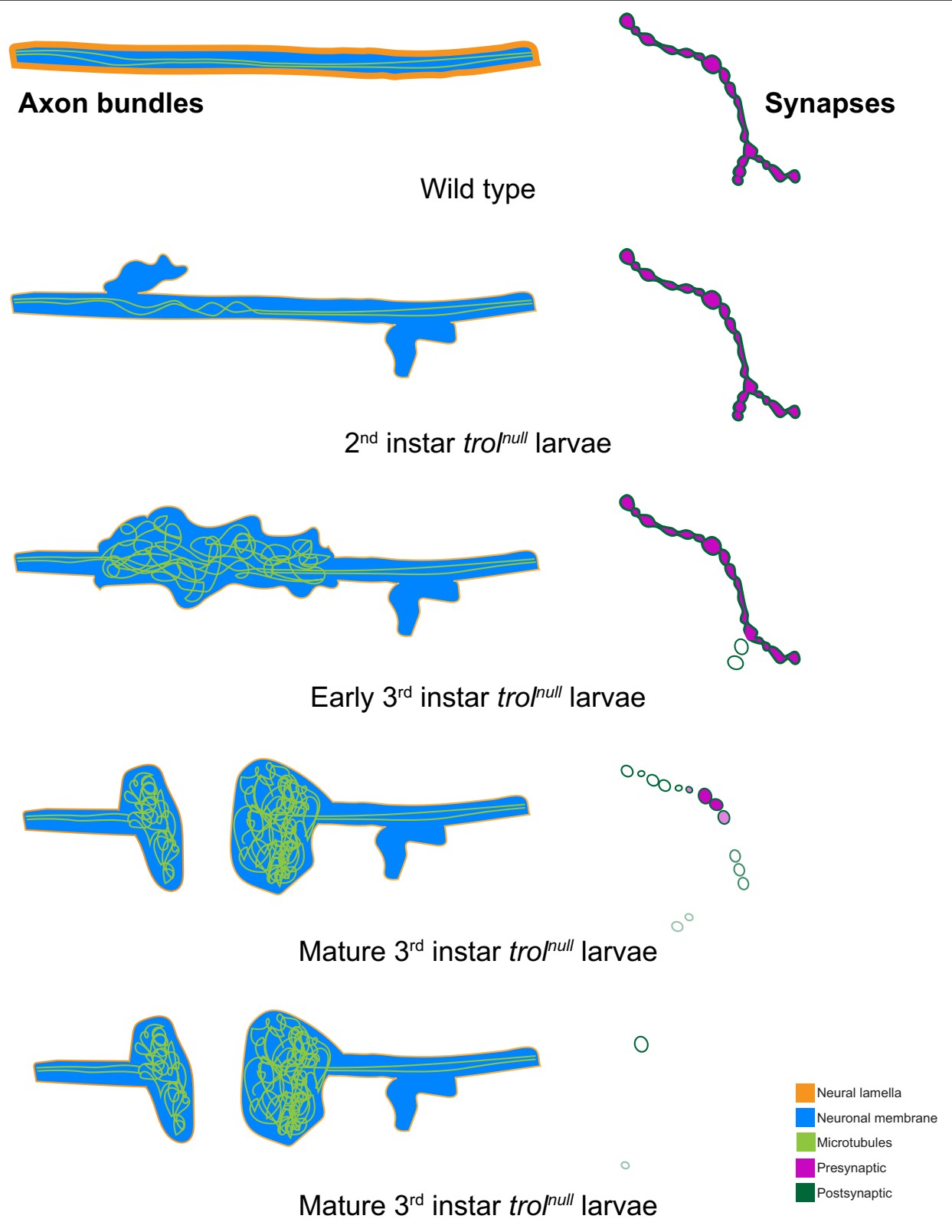

**Figure 10.** Model of progressive axonal and synaptic defects in the absence of Perlecan. In control, axonal bundles (neuronal membrane in cyan) have a thick neural lamella (orange) and continuous, straight tracks of microtubules (light green). Synapses have apposed pre- (magenta) and postsynaptic (dark green) material. In 2nd instar *trol^null* larvae, microtubules appear twisted and small protrusions of neuronal membrane and a thinner neural lamella are found. In early 3rd instar larvae lacking Perlecan, microtubules are severely disorganized and neural lamella and axonal membrane protrusions are

*Figure 10 continued on next page*

*Figure 10 continued*

larger. Some presynaptic material is lost from synapses in the first stages of retraction. By the mature 3rd instar larval stage, axons have broken entirely, and tangled nets of microtubules are observed on both sides of the breakage. Synapses are retracting and continue to retract until few or no boutons remain.

and functional defects due to disruption of the neural lamella that contributes to axonal injury. Both glia and the neural lamella can signal or interact directly with MNs (*Bittern et al., 2021*; *Edwards et al., 1993*; *Hunter et al., 2020*; *Lee and Sun, 2015*; *Meyer et al., 2014*; *Stork et al., 2008*; *Weiss et al., 2022*), providing multiple avenues by which loss of Perlecan could disrupt axonal integrity. Disruptions of interactions that Perlecan normally makes with components regulating the spectrin cytoskeleton could also contribute, as α- and β-Spectrin localize to MN axons and RNAi against either protein results in synaptic retraction (*Pielage et al., 2005*). Integrins function as ECM receptors in *Drosophila* (*Bittern et al., 2021*; *Hunter et al., 2020*) and provide a direct link between Perlecan and the spectrin cytoskeleton. Given hypomorphic Perlecan mutations cause Schwartz-Jampel syndrome in humans, a condition characterized by persistent muscle contraction and cartilage and bone abnormalities (*Arikawa-Hirasawa et al., 2002*; *Farach-Carson et al., 2014*; *Nicole et al., 2000*), and null alleles of Perlecan are incompatible with life (*Arikawa-Hirasawa et al., 1999*; *Costell et al., 1999*; *Farach-Carson et al., 2014*; *Gubbiotti et al., 2017*; *Hayes et al., 2022*), further characterization of the role of this critical ECM protein in axonal and synaptic maintenance will provide further insights into its role in tissue integrity.

## Materials and methods

### *Drosophila* stocks

Flies were maintained at 18–25°C and cultured on standard medium. 3rd instar larvae were used for all experiments unless otherwise noted. Larvae lacking Perlecan were collected at 1st or 2nd instar stage and separated from wild type counterparts and placed on petri dishes containing standard medium to facilitate survival to 3rd instar stage. Male and female larvae were used depending upon genetic background; see figure legends for genotypes. Canton-S (CS) was used as the wild type background in heterozygous controls except in the case of *Df(1)ED411/+*, where the wild type background was $w^{1118}$ (BDSC# 3605). *trol* strains used include $trol^{null}$ (*Voigt et al., 2002*; provided by Brian Stramer and Yutaka Matsubayashi), GFP-tagged *trol* (referred to in text as Trol$^{GFP}$, KDSC #110–836), *trol* deficiency (*Df(1)ED411*, BDSC #8031, 36516), *trol* overexpression constructs (UAS-*trol*.RD BDSC #65274, UAS-*trol*.RG BDSC #65273), and *trol* RNAis (UAS-*trol*-RNAi.1 VDRC #22642, UAS-*trol*-RNAi.2 VDRC #24549). Gal4 drivers used were: *tubulin*-Gal4 (BDSC #5138), *vGlut*-Gal4 (provided by Aaron DiAntonio), *elav*$^{C155}$-Gal4 (BDSC #8765), *mef2*-Gal4 (BDSC #27390), *repo*-Gal4 (*Lee and Jones, 2005*), *ppl*-Gal4 (BDSC #58768), *Lsp*-Gal4 (BDSC #6357), *Hml*-Gal4 (BDSC #6395), *c564*-Gal4 (BDSC #6982), and MN1-Ib-Gal4 (BDSC #40701). Overexpression of constitutively active Shaggy was performed using UAS-*sgg*$^{S9A}$ (BDSC #5255). Presynaptic labeling for intravital imaging was performed using CRISPR-generated nSyb$^{GFP}$ (*Guan et al., 2020*). PSD labeling for intravital imaging was performed using GluRI-IA-RFP inserted onto chromosome III under the control of its endogenous promoter (provided by Stephan Sigrist). The hypercontractive *Mhc*$^{S1}$ mutation (*Montana and Littleton, 2004*) was used to assess mechanical stress. UAS-*dSarm*-RNAi (VDRC #105369, provided by Aaron DiAntonio) was used to inhibit Wallerian degeneration in the $trol^{null}$ background. UAS-*10xGFP* (*Poukkula et al., 2011*), UAS-*CD4-TdTomato* (BDSC #77139), and UAS-*myrRFP* (BDSC #7118) were used to visualize individual motoneuron axon anatomy. Viking$^{GFP}$ (DGRC #110692, G00454 original FlyTrap identifier, provided by David Bilder) was used to visualize the neural lamella surrounding peripheral nerves.

### Immunocytochemistry

3rd instar larvae were filleted in Ca$^{2+}$-free HL3.1 solution (in mM: 70 NaCl, 5 KCl, 4 MgCl$_2$, 10 NaHCO$_3$, 5 trehalose, 115 sucrose, 5 HEPES, pH 7.18) and fixed in 4% paraformaldehyde for 15 min, washed in Ca$^{2+}$-free HL3.1 twice and 0.1-PBT (1 x PBS with 0.1% Triton X-100) once, then blocked in 5% normal goat serum (NGS) in 0.5-PBT (1 x PBS with 0.5% Triton X-100) for 30 min at room temperature or overnight at 4 °C. Samples were incubated overnight at 4 °C in blocking solution containing primary antibodies, and then washed three times with 0.1-PBT. Samples were incubated for 2 hr at room

temperature in blocking solution containing fluorophore-conjugated secondary antibodies. Primary antibodies used in this study were mouse anti-Brp at 1:500 (Nc82 DSHB, Iowa City, IA), rabbit anti-GluRIIC at 1:2000 (*Jorquera et al., 2012*), rabbit anti-Cpx at 1:5000 (*Huntwork and Littleton, 2007*), mouse anti-Dlg at 1:500 (4F3 DSHB, Iowa City, IA), mouse anti-Futsch (22C10 DSHB, Iowa City, IA) and mouse anti-GFP at 1:1000 (#A-11120, Thermo Fisher Scientific, Waltham, MA). Secondary antibodies used in this study were goat anti-mouse Alexa Fluor 488-conjugated IgG at 1:500 (#A-32723, Thermo Fisher Scientific, Waltham, MA), goat anti-rabbit Alexa Fluor 568-conjugated IgG at 1:500 (#A-11011, Thermo Fisher Scientific, Waltham, MA), goat anti-mouse 555-conjugated IgG at 1:500 (#A-32727, Thermo Fisher Scientific, Waltham, MA) and goat anti-mouse 647-conjugated IgG at 1:500 (#A-32728, Thermo Fisher Scientific, Waltham, MA). For Hrp staining, samples were incubated in DyLight 649 or 488 conjugated Hrp at 1:500 (#123-605-021, #123-485-021; Jackson ImmunoResearch Laboratories, West Grove, PA, USA). For Phalloidin staining, samples were incubated in Texas Red-X Phalloidin at 1:500 (#T7471, Thermo Fisher Scientific, Waltham, MA). Samples were mounted in Vectashield Vibrance hard setting antifade mounting medium (#H-1700, Vector Laboratories, Burlingame, CA).

## Confocal imaging and imaging data analysis

Images of fixed NMJs were acquired on a Zeiss Pascal confocal microscope (Carl Zeiss Microscopy, Zena, Germany) using a 63 X Zeiss pan-APOCHROMAT oil-immersion objective with a 1.3 NA. 3D image stacks were merged into a maximum intensity projection using Zen (Zeiss) software. Abdominal segments and muscle numbers imaged are listed in figure legends or in results text. Boutons were counted manually using the parameters specified in each experiment (e.g. Dlg +or Hrp/GluRIIC+). Severe retraction was calculated by taking the mean bouton number of control genotype(s) in an individual experiment and calculating the standard deviation. A bouton number of <mean – 2*SD and presence of postsynaptic footprints indicated severe retraction. Line profiles of fluorescence intensity across axons or boutons were generated in Volocity 3D Image Analysis software (PerkinElmer) using the 'measure line profile' algorithm in Volocity 3.2 or 5 software. Lines were drawn between 4.17–4.22 µM for axon line profiles and 2.4–2.6 µM boutons were chosen and lines extended 2 µM in either direction for bouton line profiles. Mean Vkg and Hrp intensity were calculated from maximum intensity projections using Volocity 3D Image Analysis software with the "Find Objects" algorithm and a Vkg threshold that identified neural lamella area in both genotypes. Images of live NMJs and live and fixed axons were acquired on a Zeiss Axio Imager 2 with a spinning-disk confocal head (CSU-X1; Yokagawa) and ImagEM X2 EM-CCD camera (Hamamatsu) using an Olympus LUMFL N 60 X water-immersion objective with a 1.10 NA. Low magnification images of fixed larvae were acquired on a Zeiss LSM 800 confocal microscope using a 10 x objective. A 3D image stack was acquired for each axon, NMJ, or larvae imaged. Futsch nets were counted manually. Mean fluorescence intensity of Futsch signal was calculated from the maximum intensity projection using Volocity 3D Image Analysis software. A 5 µm$^2$ ROI at the point of individual axon exit from the nerve bundle (muscle branch point) was generated and Volocity calculated the mean intensity within this ROI.

## Live intravital imaging and data analysis

Larvae were anesthetized with SUPRANE (desflurane, USP) from Amerinet Choice. Larvae were covered with a thin layer of halocarbon oil and incubated with a small paper towel containing Suprane for 1–2 min in a fume hood. Anesthetized larvae were arranged ventral side up on a glass slide between spacers made by tape. Larvae were covered with a fresh thin film of halocarbon oil and then with a cover glass. After each imaging session, larvae were placed in numbered chambers with food at room temperature. The same data acquisition settings were used to visualize NMJs at each session. Larvae were imaged every 24 hr. Area of pre- and postsynaptic material was calculated from maximum intensity projections using Volocity 3D Image Analysis software with the 'Find Objects' algorithm and a pre- or postsynaptic threshold that identified presynaptic boutons or postsynaptic receptor fields in both genotypes. The same threshold was used to analyze images from each day of analysis.

## Two-electrode voltage clamp electrophysiology and post-hoc imaging

Postsynaptic currents were recorded with a –80 mV holding potential. Experiments were performed in room temperature HL3.1 saline solution as previously described with final [Ca$^{2+}$] adjusted to 0.3 mM (*Jorquera et al., 2012*). Recordings were performed at muscle 6 of segments A3 and A4 in 3$^{rd}$ instar

larvae. Motor axon bundles were cut and individual bundles were suctioned into a glass electrode for stimulation. Action potentials were stimulated at 0.33 Hz using a programmable stimulator (Master8, AMPI; Jerusalem, Israel). Data acquisition and analysis was performed using Axoscope 9.0 and Clampfit 9.0 software (Molecular Devices, Sunnyvale, CA, USA). Inward currents are labeled on a reverse axis. For post-hoc Hrp staining, samples were incubated for 5 min in DyLight 488 conjugated Hrp (2 μL of stock solution applied directly to filleted larvae and washed twice with HL3.1 before imaging) (#123-485-021; Jackson ImmunoResearch Laboratories, West Grove, PA, USA). Live imaging was performed as described above.

## Bioinformatics

NCBI BLAST and NCBI Gene search were used to identify Perlecan homologs and to select the longest isoforms of Perlecan, Agrin, and Carrier of Wingless (Cow) in the genomes of *D. melanogaster*, *C. elegans*, *C. intestinalis*, *D. rerio*, *M. musculus*, *H. sapiens*, and *T. adhaerens*. Clustal Omega multiple sequence alignment (with default parameters) was used to align all of the sourced sequences (*Madeira et al., 2022*). Jalview was used to generate a phylogenetic tree using a BLOSUM62 matrix and average distance clustering. Sequences used for alignment and phylogenetic tree included:

| Protein | Species | Accession number (NCBI) |
|---|---|---|
| Carrier of Wingless | *D. melanogaster* | NP_001262857.1 |
| Agrin | *H. sapiens* | XP_005244806.1 |
| | *M. musculus* | XP_006538554.1 |
| | *C. intestinalis* | XP_026691460.1 |
| | *D. rerio* | XP_021325505.1 |
| | *C. elegans* | NP_001022152.3 |
| | *T. adhaerens* | XP_002113830.1 |
| Perlecan | *H. sapiens* | XP_011539620.1 |
| | *M. musculus* | XP_030109089.1 |
| | *C. intestinalis* | XP_018666843.1 |
| | *D. rerio* | XP_021325650.1 |
| | *D. melanogaster* | NP_001027034.2 |
| | *C. elegans* | NP_001364664.1 |
| | *T. adhaerens* | AKE31564.1 |

## Statistical analysis

Graphing and statistical analysis were performed using GraphPad Prism (San Diego, CA, USA).

For comparisons between two groups, statistical significance was determined using a Student's t-test. For comparisons between three of more groups, statistical significance was determined using a one-way ANOVA followed by multiple comparisons with p-values corrected for multiple hypothesis testing using either Tukey, Šidák, or Dunnett's multiple comparisons tests (individual test chosen based on Prism recommendation). Figures depict the mean of each distribution and individual data points. *N* indicates the number of individual NMJs analyzed unless otherwise noted. Number of larvae per group, mean ± SEM, *n*, and the p values are indicated in figure legends. Asterisks in the figures denote p-values of: *, $p \leq 0.05$; **, $p \leq 0.01$; ***, $p \leq 0.001$; and ****, $p \leq 0.0001$. The Source Data and Statistical Analysis file contains spreadsheets for each figure and includes all primary source data and statistical comparisons.

## Acknowledgements

This work was supported by NIH grants MH104536 and NS117588 to J.T.L. E.J.G. was supported in part by NIH pre-doctoral training grant T32GM007287 and F31NS127420. We thank the Bloomington *Drosophila* Stock Center (NIH P40OD018537), the Developmental Studies Hybridoma Bank, the Kyoto *Drosophila* Stock Center, the Vienna *Drosophila* Resource Center, Ethan Graf, Aaron DiAntonio (Washington University), Brian Stramer and Yutaka Matsubayashi (King's College London), and David Bilder (University of California Berkeley) for providing *Drosophila* strains and antibodies. We thank members of the Littleton lab for helpful discussions and comments on the project and the manuscript, especially Chad Sauvola and Patricia Pujols Vázquez.

## Additional information

### Funding

| Funder | Grant reference number | Author |
| --- | --- | --- |
| National Institute of Mental Health | MH104536 | J Troy Littleton |
| National Institute of Neurological Disorders and Stroke | NS117588 | J Troy Littleton |

The funders had no role in study design, data collection and interpretation, or the decision to submit the work for publication.

### Author contributions

Ellen J Guss, Conceptualization, Data curation, Formal analysis, Investigation, Writing - original draft, Writing - review and editing; Yulia Akbergenova, Karen L Cunningham, Data curation, Formal analysis, Writing - review and editing; J Troy Littleton, Conceptualization, Funding acquisition, Writing - original draft, Project administration, Writing - review and editing

### Author ORCIDs

J Troy Littleton http://orcid.org/0000-0001-5576-2887

Reviewer #1 (Public Review): https://doi.org/10.7554/eLife.88273.2.sa1
Reviewer #2 (Public Review): https://doi.org/10.7554/eLife.88273.2.sa2
Reviewer #3 (Public Review): https://doi.org/10.7554/eLife.88273.2.sa3
Author Response https://doi.org/10.7554/eLife.88273.2.sa4

### Data availability

All data generated or analysed during this study are included in the manuscript and supporting file; Source Data files have been provided for the figures.

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
