## [Editor Report · eLife assessment]

This study presents **valuable** new insights into the role of the extracellular matrix component (ECM) Perlecan in axon integrity, with downstream consequences for the maintenance of synaptic structures. The evidence for Perlecan's role in this process is **solid**, although negative results for Perlecan's mechanism of action should be strengthened with the addition of appropriate controls centered on the relevant pathways and mechanisms involved as well as more careful analyses and interpretations. The authors provide **convincing** data identifying and describing the cellular sequence from ECM perturbations to axonal and synaptic degeneration, but additional data pinpointing the requirements of Perlecan for axonal maintenance would further improve the impact of this study.

---

## [Referee Report · Reviewer #1 (Public Review)]

In this study Guss and colleagues identify a requirement of the ECM component Perlecan for the maintenance of neuronal structures. The authors convincingly demonstrate that the absence of Perlecan (in the entire organism) causes a severe perturbation of the ECM-based neural lamella, a support structure surrounding axon bundles and, to a lesser extent, the neuromuscular junction (NMJ). Likely because of these ECM perturbations, axons and even entire nerve bundles break at sites prior to the innervation of the peripheral muscles. Within hemisegments all affected motoneurons show signs of degeneration and synapses are retracted (degenerate). Through targeted genetic approaches in combination with immunohistochemical and electrophysiological approaches the authors aim to elucidate cell specific requirements of Perlecan. Interestingly, knock down of Perlecan in single tissues but also in combinations of tissues (neurons, glia and muscles) was not sufficient to recapitulate the phenotypes observed after ubiquitous knock down. Similarly, a rescue of these phenotypes via motoneuron expression in null mutants was not successful.

The authors very convincingly demonstrate that in the absence of Perlecan synaptic terminals degenerate and that axon and neural lamella morphology and structure is perturbed. All processes were analyzed using multiple and complementary approaches including live-imaging and electrophysiology. The precise correlation of these phenotypes and especially the careful classification into degenerated and non-affected NMJs revealed that the cause for all phenotypes is likely the disruption of the neural lamella that - through thus far unknown mechanisms - cause axonal breakage and subsequently synaptic retractions.

This study highlights the importance of the ECM to maintain neuronal structures, however, the precise source of Perlecan and the precise cause of axonal breakage remains still unresolved.

Further rescue experiments would be necessary to resolve the source of Perlecan. This requires a first demonstration that a rescue is possible with the available tools using a ubiquitous-expression analogous to the RNAi-experiments.

In addition, a careful longitudinal analysis of the integrity of individual axons (e.g. MN1 or MN4) combined with an ECM analysis may provide insights into the place and cause of the axonal breakages that are likely causal for all other observed phenotypes. As pointed out in the discussion a disruption of the blood-brain barrier at specific (?) vulnerable sites seems currently the most reasonable explanation for the observed effects. Surprisingly, the authors did not observe any rescue effect after the inhibition of Wallarian degeneration mechanisms highlighting that the cellular mechanisms underlying these two forms of degeneration in which axons are disrupted may be different.

---

## [Referee Report · Reviewer #2 (Public Review)]

In recent years, the role of the ECM in synaptic organization has been increasingly studied, leading to a better appreciation of how proteins that comprise the ECM influence synaptic structure and function. How the ECM affects neuronal structure and axonal biology is less well understood, however. Guss and colleagues begin to remedy this by assessing the role of Perlecan in the maintenance of NMJ terminals in the fly. They demonstrate a role for Perlecan in synaptic NMJ stability - loss of Perlecan results in a drastic increase in synaptic retractions. These retractions occur as a result of multiple non-cell-autonomous sources of Perlecan, as neither one tissue RNAi induces phenotypes nor does neuronal cDNA rescue a mutant. They advocate that multiple cellular mechanisms, including Wallerian degeneration and Wnt signaling, are not involved and demonstrate cytoskeletal and functional deficits. They also show that entire nerve bundles degenerate in a coordinated manner, likely due to the disruption of the neural lamella.

This is a strong and thorough genetic analysis of the role of Perlecan in neuronal stability and axonal retraction. The conclusions are largely valid, and the controls and experiments reasonable to answer the stated questions. I have some requests for additional experiments to bolster the existing conclusions.

---

## [Referee Report · Reviewer #3 (Public Review)]

The manuscript by Guss et al. characterizes an extracellular matrix protein, Perlecan (trol), in maintaining axon and synapse stability in motor neurons through its function in maintaining the neural lamella's integrity in *Drosophila*. Using a combination of immunostaining and protein labeling with fluorescent tags, the authors find that perlecan localizes to the neural lamella. When perlecan is deleted, the authors identify a synapse retraction phenotype as the subsequent result of axon damage. They further suggest that this axon instability is the result of loss of perlecan causing a disruption in the neural lamella, due to the mislocalization of neural lamella protein, Collagen IV (Vkg). Moreover, they find that perlecan acts independently of previously characterized interactions with the wnt signaling and Wallerian degradation pathways, however important controls for these negative results are lacking.

The manuscript offers an interesting and important role for perlecan in motor neuron axon maintenance. However, the experiments attempting to elucidate the mechanism of action of this protein require further validation and clarification.

---

## [Author Response]

**Reviewer 1:**

The reviewer indicated the data convincingly demonstrates absence of Perlecan causes a severe perturbation of the ECM-based neural lamella, that synaptic terminals degenerate, and that axons and even entire nerve bundles break. The reviewer noted that future studies will be important to define the precise source of Perlecan and the underlying mechanism for axonal breakage, and suggested several follow-up experiments. We address these comments below.

1. The reviewer noted our data indicate Perlecan’s role in synaptic retraction is not due to its absence from neurons and that some of the wording is confusing in this regard.

We’ve tried to make it clear throughout the manuscript that Perlecan functions non-cell autonomously, as our failure to rescue with neuronal re-expression or recapitulate the phenotype with neuronal-only RNAi indicates. As such, we agree that the phenotypes are not due to Perlecan loss within neurons, consistent with our data showing breakdown of the neural lamella ECM and subsequent axonal breakage. These phenotypes do manifest in neurons, but the defect is triggered non-cell autonomously as described in our study and stated by the reviewer here.

1. The reviewer suggested future experiments to resolve the source(s) of Perlecan secretion from defined tissues that control neuronal stability, noting that showing ubiquitous rescue with a pan-cellular Gal4 driver would be useful.

We did do pan-cellular rescue and overexpression experiments with the ubiquitous Tubulin-Gal4 driver, but expression of our two UAS-trol transgenes with this strong driver resulted in lethality. This observation indicates too much Perlecan expression is also detrimental for ECM function. Interestingly, we found that NMJ synapses do not retract following ubiquitous Perlecan overexpression in wildtype larvae, so another aspect of ECM dysfunction is responsible for lethality under this condition. As reported in the manuscript, we found driving a Trol RNAi with multiple Gal4 lines expressed in specific cell populations was unable to recapitulate the synaptic retraction phenotype, including pan-neuronal (elavC155), neuronal and muscle (elavC155 and mef2-Gal4), glial (repo-Gal4), fat body (ppl-Gal4, Lsp2-Gal4), hemocytes (Hml-Gal4), and fat body and hemocytes (c564-Gal4) driven expression. These data suggest Perlecan secretion is required by multiple cell types to achieve sufficient accumulation in the ECM to prevent neuronal instability.

1. The reviewer indicates future studies of the blood-brain barrier might reveal insights into the pathology and axonal breakage we observe. The reviewer also suggests we perform a detailed timeline of the axonal breakage timeline.

We agree with the reviewer that examination of the blood-brain barrier and glial dysfunction will be exciting experiments for future studies. For the phenotypic timeline, this was an important component of our study and was done in two ways and described in the manuscript. In Figure 4, we describe serial in vivo imaging of synapses with briefly anesthetized larvae over 4 full days of imaging. In Figure 9, we describe fixed imaging of larval axons at specific developmental stages (2nd, early 3rd, wandering 3rd instar). This set of experiments provided a detailed timeline for synaptic retraction and axonal breakage. As suggested, we also used single neuron drivers (MN1-Ib) to label a single motoneuron and examine axonal breakage and synaptic retraction at this scale. This data is shown in Figure 9E. Together, these experiments provided a timeline for the biology we observe – disruptions of the neural lamella ECM, disorganization of the axonal microtubule cytoskeleton, followed by axonal breakage and fragmentation (usually in a hemi-segment coordinated manner), with subsequent synaptic retraction at NMJs.

1. The reviewer indicates the final model in Figure 10 may not be fully representative.

We feel this model best describes our complete dataset on the Trol mutant. We provide evidence for each of these phenotypic events in detail in the paper. The disruptions to the neural lamella are described in Figure 8. The onset of synaptic retraction does occur in the 3rd instar stage and not the 2nd instar stage – Figure 4 shows this with serial in vivo imaging where we see normal synaptic morphology on Day 1 (2nd instar stage) and degeneration over the 3rd instar period (Days 2-4). The figure does not indicate Perlecan functions for synaptic stability by residing at the NMJ, only that synaptic retraction occurs. Indeed, as stated in the text, we argue against a role for Perlecan function directly at the NMJ for the phenotypes we describe, but rather as a downstream consequence of ECM disruption and following axonal breakage.

**Reviewer 2:**

The reviewer noted the work provided a strong and thorough genetic analysis of the role of Perlecan in neuronal stability and axonal retraction. The reviewer provided some suggestions for future experiments and requested a few clarifications.

1. The reviewer wondered whether mutations in other neural lamella components also cause synaptic retraction and potential genetic interactions between Trol and Vkg.

We agree further genetic studies of other neural lamella components will be of interest. In the case of Vkg, null mutations in the locus result in embryonic lethality, suggesting it plays a more critical role in overall ECM function. Although we did not perform genetic interaction studies between the two mutants (for example trans-heterozygotes), they have been shown to interact in multiple other contexts as described in the manuscript.

1. The reviewer noted the lack of whole animal Trol rescue.

As described in point #2 above, we did do pan-cellular rescue experiments with the ubiquitous driver Tubulin-Gal4, but driving our two UAS-trol transgenes resulted in lethality, indicating a strong-dosage sensitivity to Perlecan function.

1. The reviewer indicated the hyperactive Mhc mutant was an interesting experiment but only examines one alternative. They wondered if we could reduce muscle contraction and see if that "rescues" the trol phenotype.The Mhc1 null mutant is embryonic lethal, and the retraction phenotypes do not occur until the 3rd instar stage, so that experiment would not be possible. However, we did attempt to block muscle contraction by expressing a UAS-tetanus toxin to eliminate evoked neurotransmitter release with our MN1-Ib Gal4 driver (pan-neuronal expression of tetanus is lethal). This did not alter the synaptic retraction phenotype, but it was difficult to make strong conclusions for this experiment as the co-innervating Is motoneuron was not expressing tetanus toxin. As such, we did not include this data in the manuscript, though it does generally support the model that synaptic retraction is independent of muscle contraction and rather occurs downstream of the axonal breakage that we highlight.1. The reviewer wondered whether other Wnt signaling manipulations might be useful to test interactions with the Trol retraction phenotype.

Given we used the same Sgg-CA that was used to block the previously reported ghost bouton phenotype in Trol mutants and saw no effect on retraction, we did not feel that was a fruitful pathway to keep pushing on. Indeed, all our evidence point to a non-Wnt role, with neural lamella disruption and axonal breakage being the key insults.

**Reviewer 3:**

The reviewer indicated the work described an interesting and important role for Perlecan in motor neuron axon maintenance. The reviewer suggested experiments to elucidate the mechanism of action of Perlecan would benefit the study.

1. The reviewer indicated it would be beneficial to validate the Wnt and Wallerian degeneration transgenic lines used in the study to provide a positive control.

Our study used previously published and well-established Sarm RNAi and Sgg-CA transgenic lines (Sarm RNAi from the DiAntonio lab and Sgg-CA from Kamimura et al., 2013, via BDSC) that have been published multiple times and are well-validated in the field. These were not new lines that we generated. We also blocked Wallerian degeneration with a number of other perturbations to the pathway and did not see rescue of synaptic retraction in these cases either. Sarm is an upstream pathway component and thus the manipulation we included in the manuscript.

1. The reviewer notes similar questions on cell-autonomy that we addressed in point 2 to Reviewers 1 and 2 above.

The reviewer noted it would be helpful to show that the single cell-type RNAi experiments are working by western blotting for Perlecan. We performed a similar approach by examining knockdown of the endogenous Trol-GFP by the RNAi with immunostaining. Pan-cellular knockdown with Tubulin-Gal4 eliminates the staining (validating the RNAi line, Figure 1D-I), while knockdown with the individual drivers does not (Figure 5C-G). Although we used well-established cell-type specific Gal4 drivers that have been used to many other studies, we cannot eliminate strength of expression of the driver as an issue for failure to recapitulate the phenotypes. However, other experiments we performed and presented in the figures supports a non-cell autonomous role for Perlecan in axonal breakage and synaptic retraction.

1. The reviewer suggested a similar approach that Reviewer 2 did above in point 3 about the role of muscle contraction.

We agree eliminating muscle contraction altogether would be a nice assay for the role of mechanical stress, but we don’t have muscle specific drivers to eliminate contraction from only a single muscle (eliminating it everywhere is lethal). However, we did attempt to block muscle contraction by expressing a UAS-tetanus toxin to block evoked neurotransmitter release with our MN1-Ib Gal4 driver as described above. Future experiments with the newly described BoNT-C toxin produced by the Dickman lab might be a promising approach for a full elimination of all motoneuron release to achieve a similar effect and test in the Trol mutant.

1. The reviewer wondered what other components of the ECM are affected beyond Vkg in the Trol mutant.

This is an exciting question to pursue in future studies. Together with genetic interaction experiments with other ECM components, as well as a detailed analysis of the effects on glia that surround larval nerves, such studies will further refine mechanistic actions on how loss of Perlecan triggers axonal breakage and downstream synaptic retraction.